# Proteolytic cleavage and inactivation of the TRMT1 tRNA modification enzyme by SARS-CoV-2 main protease

Kejia Zhang[1], Patrick Eldin[2], Jessica H Ciesla[3], Laurence Briant[2], Jenna M Lentini[1], Jillian Ramos[1], Justin Cobb[1], Joshua Munger[3], Dragony Fu[1]*

[1]Department of Biology, Center for RNA Biology, University of Rochester, Rochester, United States; [2]Institut de Recherche en Infectiologie de Montpellier (IRIM), CNRS, UMR 9004, Université de Montpellier, Montpellier, France; [3]Department of Biochemistry and Biophysics, University of Rochester Medical Center, Rochester, United States

## eLife assessment

This manuscript provides **important** insights into the degradation of the host tRNA modification enzyme TRMT1 by the SARS-CoV-2 protease Nsp5 (nonstructural protein 5 or MPro). The data **convincingly** support the main conclusions of the paper. These results will be of interest to virologists studying the alterations in tRNA modifications, host methyltransferases, and viral infections.

*For correspondence:
DFU4@ur.rochester.edu

Competing interest: The authors declare that no competing interests exist.

**Abstract** Nonstructural protein 5 (Nsp5) is the main protease of SARS-CoV-2 that cleaves viral polyproteins into individual polypeptides necessary for viral replication. Here, we show that Nsp5 binds and cleaves human tRNA methyltransferase 1 (TRMT1), a host enzyme required for a prevalent post-transcriptional modification in tRNAs. Human cells infected with SARS-CoV-2 exhibit a decrease in TRMT1 protein levels and TRMT1-catalyzed tRNA modifications, consistent with TRMT1 cleavage and inactivation by Nsp5. Nsp5 cleaves TRMT1 at a specific position that matches the consensus sequence of SARS-CoV-2 polyprotein cleavage sites, and a single mutation within the sequence inhibits Nsp5-dependent proteolysis of TRMT1. The TRMT1 cleavage fragments exhibit altered RNA binding activity and are unable to rescue tRNA modification in TRMT1-deficient human cells. Compared to wild-type human cells, TRMT1-deficient human cells infected with SARS-CoV-2 exhibit reduced levels of intracellular viral RNA. These findings provide evidence that Nsp5-dependent cleavage of TRMT1 and perturbation of tRNA modification patterns contribute to the cellular pathogenesis of SARS-CoV-2 infection.

## Introduction

Severe acute respiratory syndrome coronavirus 2 (SARS-CoV-2) is an enveloped, single-stranded RNA virus that is the causative agent of the COVID-19 pandemic (*Lamers and Haagmans, 2022*; *Merad et al., 2022*). SARS-CoV-2 is a member of the Betacoronavirus genus of the Coronaviridae family, which includes the severe acute respiratory syndrome coronavirus 1 (SARS-CoV-1) and Middle East respiratory syndrome coronavirus (MERS-CoV) (reviewed in *Hartenian et al., 2020*). SARS-CoV-2 primarily targets the human respiratory tract and lungs, which can clinically manifest as an acute respiratory distress syndrome. Once delivered into the host cell, the incoming positive-strand viral RNA genome is first translated by host ribosomes into two overlapping polyproteins, pp1a and pp1ab. Viral polyproteins are then proteolytically processed into 16 mature nonstructural proteins (Nsps) involved

**eLife digest** The virus responsible for COVID-19 infections is known as SARS-CoV-2. Like all viruses, SARS-CoV-2 carries instructions to make proteins and other molecules that play essential roles in enabling the virus to multiply and spread. Viruses are unable to make these molecules themselves, so they infect cells and trick them into making the molecules and assembling new virus particles on their behalf instead.

When SARS-CoV2 infects cells, the host cells are reprogrammed to make chains containing several virus proteins that need to be severed from each other by a virus enzyme, known as Nsp5, to enable the proteins to work properly. Previous studies suggested that Nsp5 may also interact with a human protein known as TRMT1, which helps with the production of new proteins in cells. However, it was not clear how Nsp5 may bind to TRMT1 or how this interaction may affect the host cell.

Zhang et al. used biochemical and molecular techniques in human cells to study how Nsp5 interacts with TRMT1. The experiments found that the virus enzyme cuts TRMT1 into fragments that are inactive and are subsequently destroyed by the cells. Moreover, Nsp5 cuts TRMT1 at exactly the same position corresponding to the cleavage sites of the viral proteins. Mutation of the sequence in TRMT1 renders Nsp5 ineffective at cutting the protein.

SARS-CoV-2 infection caused TRMT1 levels to decrease inside the cells, in turn, leading to a drop in TRMT1 activity. The virus multiplied less in cells that were unable to produce TRMT1 compared to normal human cells, suggesting that the virus benefits from TRMT1 early during infection, before inactivating it at a later point.

These findings suggest that one way SARS-CoV-2 causes disease is by decreasing the levels of a human protein that regulates protein production. In the future, the work of Zhang et al. may provide new markers for detecting infections of SARS-CoV-2 and other similar viruses and guide efforts to make more effective therapies against them.

in the assembly of the viral replication-transcription complex (reviewed in *V'kovski et al., 2021*). The Nsps also participate in disrupting host cellular processes and pathways to escape immune recognition and facilitate viral propagation (reviewed in *Minkoff and tenOever, 2023*; *Suryawanshi et al., 2021*).

The maturation step to release the individual Nsp polypeptides is executed by two viral-encoded proteases: Nsp5 (also known as Main Protease, M^Pro/3C-like protease) and Nsp3 (also known as Papain-Like Protease, PL^Pro) (*Narayanan et al., 2022*). Nsp3 is contained within pp1a and releases itself from the polyprotein through self-cleavage. Nsp3 also cleaves pp1a at multiple sites to release Nsp1 through Nsp4 (*Yan and Wu, 2021*). The Nsp5 main protease processes pp1b at eleven sites to release itself and Nsp4 to Nsp16 (*Chen et al., 2021*). Besides cleaving viral substrates, Nsp3 and Nsp5 have also been found to cleave endogenous host proteins linked to the immune response and cell survival (*Liu et al., 2021*; *Meyer et al., 2021*; *Meyers et al., 2021*; *Moustaqil et al., 2021*; *Wenzel et al., 2021*; *Zhang et al., 2021b*). Nsp3 and Nsp5 are essential for viral replication and represent well-characterized drug targets among coronaviruses. Notably, the active component of the antiviral drug Paxlovid is nirmatrelvir, a reversible covalent inhibitor of the Nsp5 main protease (*Owen et al., 2021*). By inhibiting Nsp5 proteolytic activity, Paxlovid reduces viral replication and disease severity in patients with COVID-19.

Purification of individual SARS-CoV-2 proteins from human cells have identified a potential interaction between a catalytic-inactive version of Nsp5 with human TRMT1 (*Gordon et al., 2020a*). A cross-coronavirus investigation from the same group has extended this observation by finding that TRMT1 interacts with a catalytic-inactive version of Nsp5 encoded by SARS-CoV-1, which is 98.7% similar to SARS-CoV-2 (*Gordon et al., 2020b*). Interestingly, the same study found no detectable interaction between TRMT1 and Nsp5 encoded by MERS-CoV, which has 66.8% similarity with SARS-CoV-2. TRMT1 was also identified as a potential interactor with catalytic-inactive Nsp5 using proximity-dependent biotinylation (*Samavarchi-Tehrani et al., 2020*).

TRMT1 is a tRNA modification enzyme that catalyzes the formation of dimethylguanosine (m2,2G) at position 26 in more than half of all human tRNAs (*Dewe et al., 2017*; *Jonkhout et al., 2021*). The m2,2G modification is located at a key structural position in tRNAs and is hypothesized to play a role in

proper tRNA folding (*Pallan et al., 2008*; *Steinberg and Cedergren, 1995*). TRMT1-deficient human cells exhibit perturbations in global protein synthesis and decreased proliferation (*Dewe et al., 2017*). Moreover, loss-of-function variants in the *TRMT1* gene are the cause of certain types of intellectual disability disorders in humans (*Blaesius et al., 2018*; *Najmabadi et al., 2011*; *Zhang et al., 2020*). Thus, changes in TRMT1 activity can impact protein synthesis leading to downstream cellular and biological effects.

The life cycle of many viruses has been linked to host tRNA biology (reviewed in *Dremel et al., 2023*; *Nunes et al., 2020*). In one well-known example, retroviruses such as HIV use cellular tRNAs as primers for reverse transcription and other viral functions (reviewed in *Jin and Musier-Forsyth, 2019*). DNA and RNA viruses can also impact tRNA synthesis, processing, and charging to modulate infection and pathogenesis (*Dremel et al., 2022*; *Netzer et al., 2009*; *Tucker et al., 2020*). In a recent study, Chikungunya RNA virus infection has been shown to increase the expression of a human tRNA modification enzyme and alter tRNA modification patterns to favor viral protein expression (*Jung-fleisch et al., 2022*). Interestingly, SARS-CoV-2 viral particles exhibit an enrichment of specific host tRNAs with distinct modification profiles (*Peña et al., 2022*). Moreover, analysis of RNA extracted from human nose swabs have identified tRNA profiles and modification signatures associated with mild versus severe COVID-19 (*Katanski et al., 2022*). The change in tRNA modification profiles could represent another mechanism employed by SARS-CoV-2 to take over translation in addition to previously described strategies (*Finkel et al., 2021*).

The putative interaction between Nsp5 and TRMT1 indicates that SARS-CoV-2 infection could affect the function of TRMT1 with consequent effects on tRNA modification levels. However, the association between Nsp5 and TRMT1 has not been validated nor characterized. Here, we find that TRMT1 is an endogenous cleavage target of Nsp5 resulting in methyltransferase inactive cleavage products. Moreover, we find that SARS-CoV-2 infection correlates with decreased TRMT1 levels and a reduction in m2,2G-modified tRNAs. In addition, we provide evidence that TRMT1 expression is required for efficient SARS-CoV-2 replication in human cells. These studies uncover TRMT1 as a novel proteolytic cleavage target of Nsp5 during SARS-CoV-2 infection and suggest possible pathological mechanisms associated with perturbations in TRMT1-catalyzed tRNA modifications.

## Results

### SARS-CoV-2 infection reduces cellular levels of TRMT1 and TRMT1-catalyzed tRNA modifications

To test the effects of SARS-CoV-2 infection on TRMT1 and tRNA modifications, we used an MRC-5 human lung fibroblast cell line expressing the ACE2 receptor that is permissive for SARS-CoV-2 infection (*Raymonda et al., 2022*; *Uemura et al., 2021a*; *Uemura et al., 2021b*). MRC-5 cells were mock-infected or infected with SARS-CoV-2 followed by harvesting at 24- and 48 hr post-infection for sample preparation. We chose the 24- and 48 hr time points since MRC5-ACE2 cells exhibit elevated accumulation of viral RNA at these time points without extreme cell death that occurs at subsequent time points (*Raymonda et al., 2022*). Immunoblotting for the SARS-CoV-2 nucleocapsid (N) protein confirmed the infection and expression of viral proteins in MRC-5 cells compared to mock-infected cells (*Figure 1A and N* protein, compare lanes 1 and 2 to lanes 3 and 4). To probe for TRMT1, we used an antibody that detects the major TRMT1 isoform of ~72 kDa as well as a non-specific band at ~65 kDa in human cell lysates (*Dewe et al., 2017*; *Perez et al., 2022*). Using this antibody, we also detected the 72 kDa TRMT1 isoform as well as the non-specific 65 kDa band in MRC-5 cell lysates (*Figure 1A*, circle denotes TRMT1, asterisk denotes non-specific band). In multiple independent replicates, MRC-5 cells infected with SARS-CoV-2 exhibited a ~30% reduction in TRMT1 levels at 24- and 48 hr post-infection compared to mock-infected cells (*Figure 1B*). These results provide evidence that SARS-CoV-2 infection reduces cellular levels of TRMT1.

We next measured the levels of TRMT1-catalyzed m2,2G modifications in cellular RNA using quantitative mass spectrometry. While m2,2G levels did not exhibit a significant change between mock-infected and SARS-CoV-2-infected cells at 24 hr post-infection, m2,2G levels were decreased by ~15% at 48 hr post-infection in multiple biological replicates (*Figure 1C*). The decrease in m2,2G levels after SARS-CoV-2 infection was also reproduced in an independent experiment (*Figure 1—figure supplement 1*). In addition to m2,2G, more than half of all tested RNA modifications exhibited a decrease

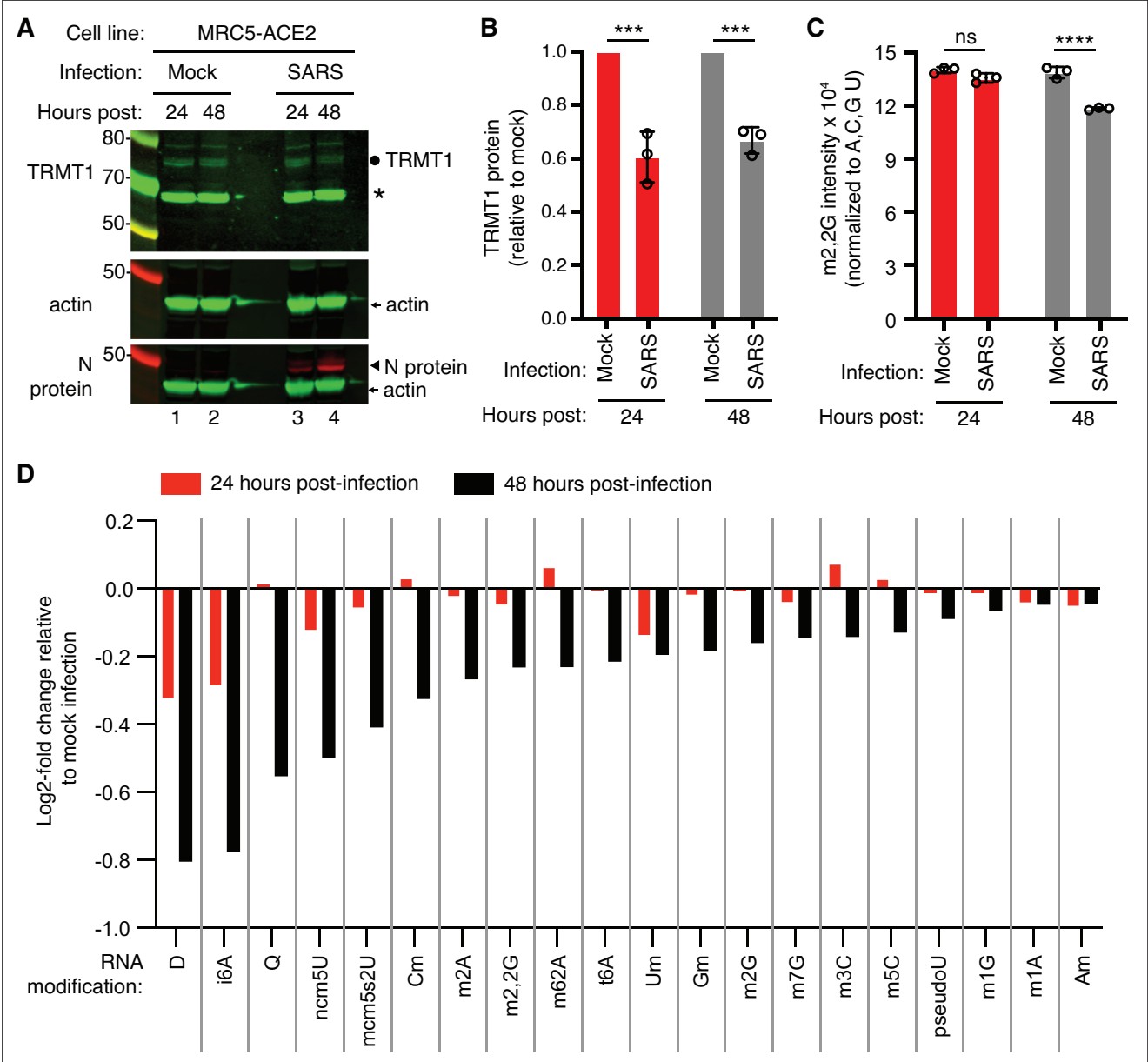

**Figure 1.** Human cells infected with severe acute respiratory syndrome coronavirus 2 (SARS-CoV-2) exhibit a reduction in tRNA methyltransferase 1 (TRMT1) levels and perturbations in tRNA modification patterns. (**A**) Immunoblot analysis of lysates prepared from MRC-5-ACE2 human cells that were mock-infected or infected with SARS-CoV-2 at multiplicity of infection (MOI) of 5 for 24 or 48 hr. The immunoblot was probed with anti-TRMT1, actin, or SARS-CoV-2 nucleocapsid (N) antibodies. Circle represents endogenous full-length TRMT1. Asterisk (*) denotes a non-specific band. Size markers are noted in kiloDalton. (**B**) Quantification of TRMT1 signal intensity normalized to actin in the mock or SARS-CoV-2-infected cell lines. TRMT1 protein levels are expressed relative to mock-infected samples for each time point. (**C**) m2,2G levels in small RNAs isolated from MRC5 cells that were either mock-infected or infected with SARS-CoV-2 at MOI of 5 for 24 or 48 hr. m2,2G levels were normalized to A, C, G, and U. Samples were measured in biological replicates. Statistical significance for (**B**) and (**C**) was determined by two-way ANOVA with multiple comparisons test. ***p<0.001; ****p<0.0001; ns, non-significant. (**D**) Levels of the indicated RNA modifications in small RNAs isolated from MRC5 cells that were either mock-infected or infected with SARS-CoV-2 for 24 or 48 hr. RNA modification levels were normalized to A, C, G, and U. Y-axis represents the log2 fold change in the levels of the indicated tRNA modification between SARS-CoV-2 infected versus mock-infected MRC5 cells. The experiment in (**C**) was repeated as an independent biological replicate in *Figure 1—figure supplement 1*.

The online version of this article includes the following source data and figure supplement(s) for figure 1:

**Source data 1.** Raw uncropped immunoblots for *Figure 1A*.

**Source data 2.** LC-MS measurements of RNA modifications.

*Figure 1 continued on next page*

*Figure 1 continued*

**Figure supplement 1.** LC-MS analysis of dimethylguanosine (m2,2G) levels in small RNAs isolated from MRC5 cells that were either mock-infected or infected with severe acute respiratory syndrome coronavirus 2 (SARS-CoV-2) at multiplicity of infection (MOI) of 5 for 24 or 48 hr.

**Figure supplement 1—source data 1.** LC-MS measurements of RNA modifications.

in SARS-CoV-2-infected cells compared to mock-infected cells at 24 hr post-infection (*Figure 1D*). At 48 hr post-infection with SARS-CoV-2, most of the tested RNA modifications exhibited a decrease in levels compared to mock-infected cells. The RNA modifications that decrease upon viral infection include modifications that are known to be present in multiple types of RNA, including tRNA, rRNA, and mRNA. Thus, SARS-CoV-2 infection of human MRC-5 lung fibroblast cells alters the steady-state levels of multiple RNA modifications, including the m2,2G modification catalyzed by TRMT1.

## Nsp5 interacts with TRMT1

Examination of the primary structure of human TRMT1 revealed an eight amino acid residue sequence between the methyltransferase domain and zinc-finger motif matching the consensus sequence of Nsp5 cleavage sites in SARS-CoV-2 polyproteins (*Figure 2A and B*). The putative cleavage site in TRMT1 includes a glutamine (Q) residue conserved at the fourth position that is found in all Nsp5 cleavage sites of SARS-CoV-2 polyproteins (*Figure 2B*). Based upon a predicted tertiary structure of TRMT1 using Alpha Fold (*Jumper et al., 2021*; *Varadi et al., 2022*), this putative Nsp5 cleavage site is expected to lie in an unstructured linker region exposed on the surface of TRMT1 (*Figure 2C*).

To investigate a potential interaction between Nsp5 and TRMT1, we used human 293T cells due to their high transfection efficiency. We transfected human 293T cells with plasmids expressing wild-type (WT) Nsp5 from SARS-CoV-2 or Nsp5-C145A. The Nsp5-C145A mutant contains an alanine substitution of the catalytic cysteine residue in the active site of Nsp5 and is proteolytically inactive (*Lee et al., 2020*). The Nsp5 proteins were expressed as fusion proteins with the Twin-Strep purification tag (*Schmidt et al., 2013*). We also tested the interaction between Strep-tagged Nsp5 or Nsp5-C145A with TRMT1 by co-expression with TRMT1 fused with a carboxyl-terminal FLAG tag. TRMT1 was tagged at the carboxy-terminus to prevent interference with the amino-terminal mitochondrial targeting signal (*Dewe et al., 2017*). The Strep-tagged Nsp5 or Nsp5-C145A mutants were affinity purified from cell lysates on strep-tactin resin, eluted with biotin, and purified proteins detected by immunoblotting. Probing of input lysates and purified samples with anti-Strep antibodies confirmed the expression and recovery of WT-Nsp5 and Nsp5-C145A (*Figure 2D*, Strep). Nsp5-C145A protein accumulated at higher levels compared to WT-Nsp5 due to the absence of self-cleavage by Nsp5-C145A and the known toxicity of expressing WT-Nsp5 in mammalian cells (*Resnick et al., 2021*; *Wenzel et al., 2021*).

To probe for TRMT1, we used the antibody described above that detects endogenous full-length TRMT1 of 72 kDa (*Figure 2D*, input lanes 1–3, circle denotes endogenous TRMT1). The antibody also detected the overexpressed TRMT1-FLAG protein (*Figure 2D*, lanes 4–6, square denotes TRMT1-FLAG). Only background levels of endogenous TRMT1 or TRMT1-FLAG were detected in the vector or WT-Nsp5 purifications (*Figure 2D*, TRMT1, low or high exposure, lanes 7, 8, 10, and 11). In contrast, endogenous TRMT1 was enriched in the purification with Nsp5-C145A (*Figure 2D*, TRMT1 high exposure, lane 9). TRMT1-FLAG was also detected specifically in the Nsp5-C145A purification compared to the control or Nsp5 purifications (*Figure 2D*, compare lanes 10 and 11 to lane 12). The association between TRMT1 and Nsp5-C145A was reproduced in an independent purification (*Figure 2—figure supplement 1*). These results provide evidence that Nsp5 can interact with TRMT1 when Nsp5 is in the catalytic-inactive form.

## Nsp5 cleaves TRMT1 in a site-specific manner

The interaction of TRMT1 with Nsp5-C145A along with cleavage site prediction suggests that TRMT1 could be a proteolysis substrate of Nsp5. To test this hypothesis, we monitored for TRMT1 cleavage in 293T human cells expressing Strep-tagged Nsp5 or GFP as a control. Cleavage after amino acid residue 530 of TRMT1 is predicted to result in an N-terminal fragment of 58 kDa and a C-terminal fragment of 14 kDa. To probe for the N-terminal TRMT1 fragment, we used the antibody described above which targets residues 201–229 of TRMT1. Using this antibody, we detected full-length TRMT1 and the ~65 kDa non-specific band in lysates from 293T cells expressing Strep-GFP (*Figure 3A*, TRMT1,

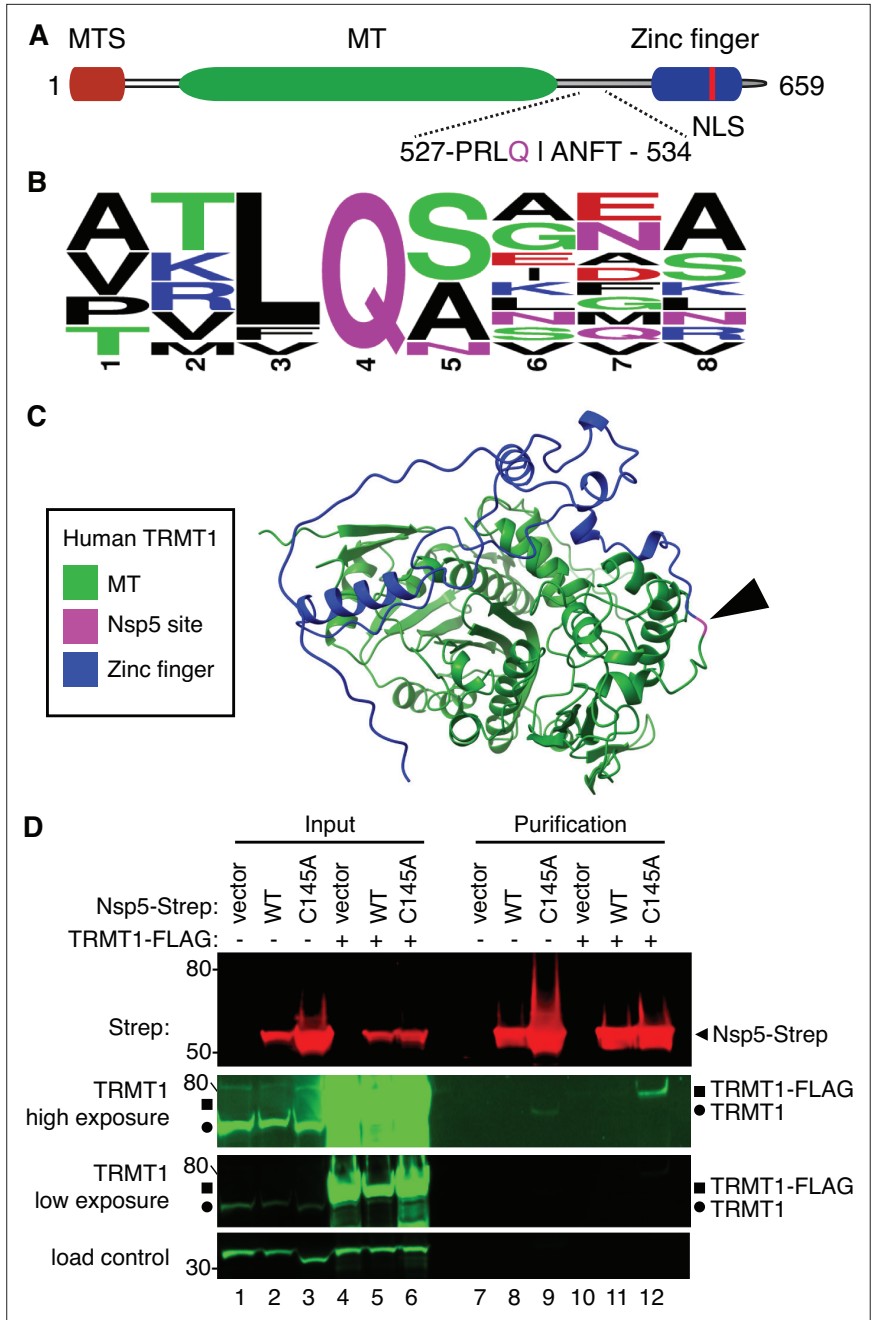

**Figure 2.** Severe acute respiratory syndrome coronavirus 2 (SARS-CoV-2) nonstructural protein 5 (Nsp5) binds tRNA methyltransferase 1 (TRMT1) in human cells. (**A**) Schematic of human TRMT1 primary structure with predicted Nsp5 cleavage site. Mitochondrial targeting signal (MTS), methyltransferase (MT) domain, and zinc finger motif are denoted. (**B**) Consensus sequence logo of cleavage sites in SARS-CoV-2 polyproteins. (**C**) Alpha-fold predicted structure of human TRMT1 with putative Nsp5 cleavage site denoted in magenta and arrowhead. (**D**) Immunoblot of input and strep-tactin purifications from human cells expressing empty vector, wild-type (WT) Nsp5, or Nsp5-C145A fused to the Strep-tag without or with co-expression with TRMT1-FLAG. The immunoblot was probed with anti-Strep, FLAG, and actin antibodies. Square represents TRMT1-FLAG, circle represents endogenous TRMT1. Size markers are noted to the left in kiloDalton. The experiment in (**D**) was repeated as an independent biological replicate in *Figure 2—figure supplement 1*.

The online version of this article includes the following source data and figure supplement(s) for figure 2:

**Source data 1.** Raw uncropped immunoblots for *Figure 2D*.

*Figure 2 continued on next page*

*Figure 2 continued*

**Figure supplement 1.** Immunoblot of input and strep-tactin purifications from human cells expressing empty vector, wild-type (WT) nonstructural protein 5 (Nsp5), or Nsp5-C145A fused to the Strep-tag without or with co-expression with tRNA methyltransferase 1 (TRMT1)-FLAG.

**Figure supplement 1—source data 1.** Raw uncropped immunoblots for *Figure 2—figure supplement 1*.

lanes 1–3, circle and asterisk, respectively). No detectable change in the levels of full-length TRMT1 was observed in human cells expressing Nsp5 or Nsp5-C145A (*Figure 3A*, TRMT1, quantified in 3B). However, lysates prepared from human cells expressing Nsp5 exhibited an additional band migrating below the non-specific band that matches the predicted size of the N-terminal TRMT1 fragment (*Figure 3*, TRMT1, lanes 4–6, arrow). The putative N-terminal TRMT1 fragment was detectable at 24 hr post-transfection with the Nsp5-expression plasmid, increased at 48 hr, and remained detectable at 72 hr post-transfection (*Figure 3B*). In contrast, the N-terminal TRMT1 fragment was not

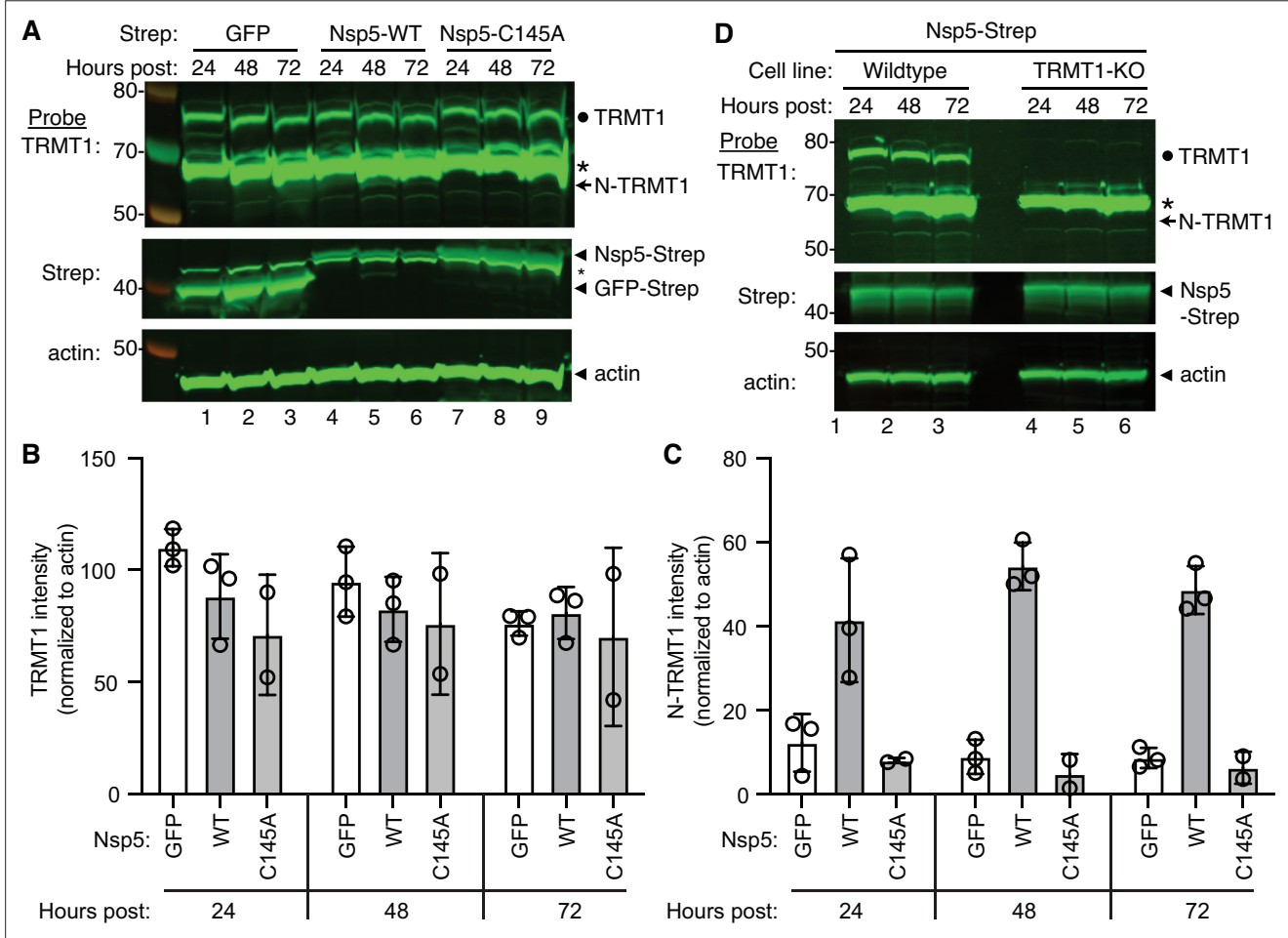

**Figure 3.** Nonstructural protein 5 (Nsp5) expression induces cleavage of tRNA methyltransferase 1 (TRMT1) in human cells. (**A**) Immunoblot of lysates prepared from human 293T cells expressing GFP, Nsp5 or Nsp5-C145A. The immunoblot was probed with anti-TRMT1, Strep, or actin antibodies. Hours post represents the time post-transfection. Circle represents endogenous TRMT1. Arrow represents the N-terminal (N)-TRMT1 cleavage fragment. Asterisk (*) denotes a non-specific band. Size markers to the left in kiloDalton. (**B, C**) Quantification of endogenous TRMT1 or N-terminal (N)-TRMT1 cleavage product in transfected cells. TRMT1 and N-TRMT1 signal was normalized to actin. (**D**) Immunoblot of lysates prepared from wild-type or TRMT1-knockout (KO) human cell lines expressing Nsp5. Experiments in (**A**) and (**D**) were repeated three times in biological replicates (see source data).

The online version of this article includes the following source data for figure 3:

**Source data 1.** Raw uncropped immunoblots for *Figure 3*.

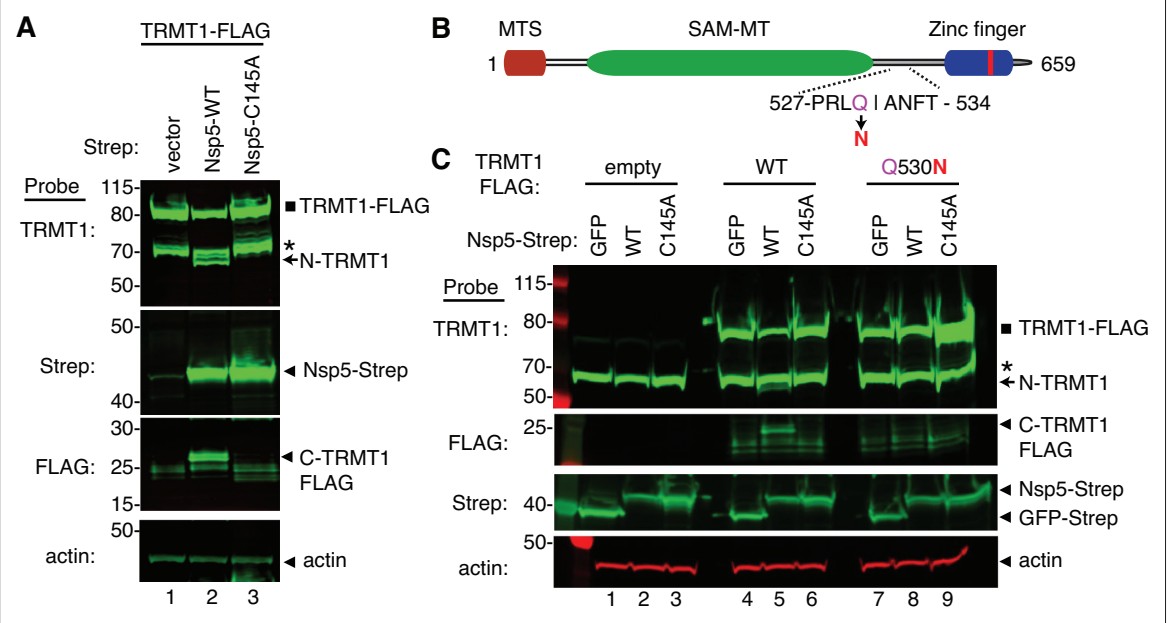

**Figure 4.** Sequence-dependent cleavage of tRNA methyltransferase 1 (TRMT1) by severe acute respiratory syndrome coronavirus 2 (SARS-CoV-2) nonstructural protein 5 (Nsp5). (**A**) Immunoblot of lysates from human cells expressing empty vector, wild-type (WT) Nsp5-Strep, or Nsp5-C145A-Strep without or with co-expression with TRMT1-FLAG. The immunoblot was probed with anti-Strep, FLAG, and actin antibodies. Square represents TRMT1-FLAG, Asterisk (*) denotes a non-specific band, arrow represents N-terminal TRMT1 cleavage product and arrowhead indicates the C-terminal TRMT1 cleavage product. (**B**) Schematic of human TRMT1 with predicted Nsp5 cleavage site and Q530N mutation. (**C**) Immunoblot of lysates from human cells expressing empty vector, wild-type (WT) Nsp5-Strep, or Nsp5-C145A-Strep without or with co-expression with TRMT1-FLAG or TRMT1-FLAG Q530N. Experiments in (**A**) and (**C**) were repeated three times as biological replicates with comparable results (see source data).

The online version of this article includes the following source data and figure supplement(s) for figure 4:

**Source data 1.** Raw uncropped immunoblots for *Figure 4*.

**Figure supplement 1.** Detection of the C-terminal tRNA methyltransferase 1 (TRMT1) fragment produced by nonstructural protein 5 (Nsp5)-dependent cleavage in human cells.

**Figure supplement 1—source data 1.** Raw uncropped immunoblots for *Figure 4—figure supplement 1*.

detected above the background signal in human cells expressing the proteolytically inactive Nsp5-C145A mutant (*Figure 3B*, TRMT1, lanes 7–9, quantified in 3 C).

We validated the specificity of our results by using a human 293T TRMT1-knockout (KO) cell line that is deficient in TRMT1 expression (*Dewe et al., 2017*; *Zhang et al., 2020*). In this case, neither full-length TRMT1 nor the TRMT1 cleavage fragment were detected in a TRMT1-deficient human cell line expressing Nsp5 (*Figure 3D*, compare lanes 1 through 3 to lanes 4 through 6). This result provides additional confirmation that the N-terminal TRMT1 fragment arises from the cleavage of endogenous TRMT1 by Nsp5.

We also attempted to detect the C-terminal TRMT1 cleavage fragment in human cells expressing Nsp5 using an antibody targeting residues 609–659 of TRMT1 (*Zhang et al., 2021c*). This anti-TRMT1 antibody detects the full-length 72 kDa TRMT1 isoform that is absent in TRMT1-KO cells (*Figure 4—figure supplement 1A*, lanes 1 and 2). However, no additional band matching the expected size of the C-terminal fragment was detected in human cells expressing Nsp5 compared to cells expressing GFP or Nsp5-C145A (*Figure 4—figure supplement 1A*, lanes 3 through 6). This could be due to degradation of the C-terminal TRMT1 fragment and/or low sensitivity of the antibody.

To increase the sensitivity for detecting the C-terminal TRMT1 fragment, we co-transfected Nsp5 expression plasmids along with a plasmid encoding TRMT1 fused to a FLAG tag at the C-terminus. Confirming the results above with endogenous TRMT1, the N-terminal TRMT1 cleavage product was detected in the lysate of human cells overexpressing TRMT1-FLAG and Nsp5 but not with vector or Nsp5-C145A (*Figure 4A*, TRMT1, compare lanes 1 and 3 to lane 2). In addition, using an antibody against the FLAG tag, we could detect a ~20 kDa product matching the expected molecular weight

of a FLAG-tagged C-terminal TRMT1 cleavage product in the lysate of human cells expressing Nsp5 but not vector alone or Nsp5-C145A (*Figure 4A*, FLAG, arrowhead). To confirm that the 20 kDa band detected with the anti-FLAG antibody was the C-terminal portion of TRMT1, we probed the cell lysates with the anti-TRMT1 antibody targeting residues 609–659 noted above. Using this antibody, we detected the same ~20 kDa band in the lysate of human cells expressing Nsp5 but not vector alone or Nsp5-C145A (*Figure 4—figure supplement 1B*, lanes 1–3). These data provide evidence that SARS-CoV-2 Nsp5 cleaves at the predicted cleavage site in TRMT1 leading to N- and C-terminal fragments in human cells.

All Nsp5 cleavage sites in SARS-CoV-1 and SARS-CoV-2 polyproteins contain a glutamine residue at position four (*Figure 2B*; *Grum-Tokars et al., 2008*; *Jin et al., 2022*; *Lee et al., 2022*). Mutation of the glutamine to asparagine is sufficient to abolish recognition and cleavage by Nsp5 from SARS-CoV-1 or SARS-CoV-2 (*Heilmann et al., 2022*; *Muramatsu et al., 2013*). Thus, we tested if TRMT1 exhibited the same requirements for cleavage by Nsp5 by generating an expression construct for TRMT1-FLAG in which residue Q530 of the predicted Nsp5 cleavage site was mutated to asparagine (*Figure 4B*, Q530N). Further confirming our results above, expression of TRMT1-FLAG with WT-Nsp5 but not GFP or Nsp5-C145A led to the accumulation of N- and C-terminal TRMT1 cleavage fragments (*Figure 4C*, TRMT1 and FLAG, lanes 4–6, arrow and arrowhead, *Figure 4—figure supplement 1B*). In contrast to wild-type TRMT1, the appearance of the N- or C-terminal TRMT1 cleavage fragments was barely detectable when the TRMT1-Q530N mutant was co-expressed with Nsp5 (*Figure 4C*, lane 8, *Figure 4—figure supplement 1B*). Altogether, these results demonstrate that TRMT1 can be recognized and cleaved by Nsp5 in human cells with cleavage requiring a sequence that matches Nsp5 cleavage sites in SARS-CoV-2 polyproteins.

## Functional properties of TRMT1 fragments resulting from Nsp5 cleavage

Cleavage of TRMT1 after residue Q530 will result in an N-terminal fragment encompassing the methyltransferase domain and a C-terminal TRMT1 fragment containing a zinc finger motif that mediates tRNA interaction (*Dewe et al., 2017*; *Zhang et al., 2020*). Thus, we tested the functional properties of the predicted TRMT1 cleavage fragments compared to full-length TRMT1. First, we tested the interaction between the TRMT1 cleavage fragments and RNA. We have previously shown that TRMT1 exhibits a stable interaction with rRNAs and substrate tRNAs that are targets for m2,2G modification (*Dewe et al., 2017*; *Zhang et al., 2020*). Based upon this interaction, we expressed full-length TRMT1, the TRMT1-Q530N mutant, or the TRMT1 fragments as FLAG-tagged fusion proteins in 293T human cells followed by affinity purification and analysis of copurifying RNAs (*Figure 5A*). Immunoblotting confirmed the expression and purification of each TRMT1 variant on antiFLAG resin (*Figure 5B*). As expected, the purification of full-length TRMT1 resulted in the enrichment of rRNA and tRNAs compared to the control purification from vector-transfected cells (*Figure 5C*, compare lane 6 to lane 7). The Q530N variant of TRMT1 also exhibited comparable binding of tRNAs as full-length TRMT1 (*Figure 5C*, lane 10). In contrast, no detectable enrichment of rRNA or tRNA was detected for the N-terminal TRMT1 fragment (*Figure 5C*, lane 8). Interestingly, the TRMT1 C-terminal fragment exhibited similar levels of tRNA binding as full-length TRMT1 (*Figure 5C*, compare lanes 7–9). Thus, the N-terminal TRMT1 cleavage fragment appears to be insufficient for binding tRNA while the C-terminal fragment containing the Zn-finger motif is sufficient for binding to tRNAs.

To further dissect the functionality of the TRMT1 cleavage fragments, we next used the TRMT1-KO human cell line described above. This TRMT1-KO cell line is deficient in TRMT1 and lacks m2,2G modifications in all tested tRNAs containing G at position 26 (*Dewe et al., 2017*; *Zhang et al., 2020*). Using transient transfection, we expressed full-length TRMT1 or the TRMT1 variants in either the control wild-type (WT) or TRMT1-KO cell lines (*Figure 5D*). We then assessed for rescue of m2,2G formation in nuclear-encoded tRNA-Met-CAU or mitochondrial-encoded (mt)-tRNA-Ile-UAU using a reverse transcriptase (RT)-based primer extension assay. Based upon this assay, vector-transfected WT human cells exhibited an RT block at position 26 of tRNA-Met-CAU and mt-tRNA-Ile-UAU indicative of the m2,2G modification (*Figure 5E*, Lane 1). No read-through product was detected for either tRNA in control human cells indicating that nearly all endogenous tRNA-Met-CAU and mt-tRNA-Ile-UAU is modified with m2,2G. Consistent with this observation, increased expression of full-length

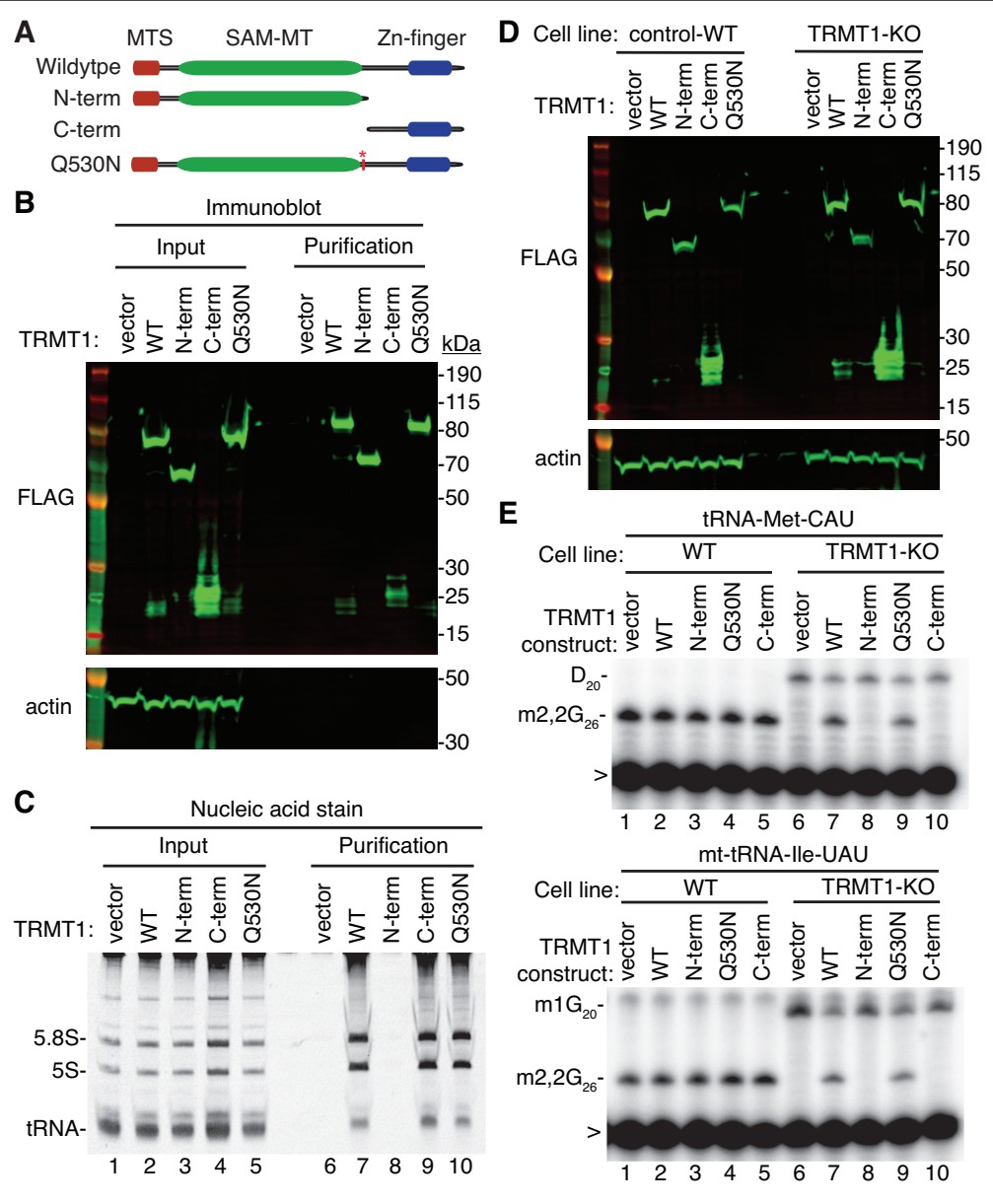

**Figure 5.** N- and C-terminal tRNA methyltransferase 1 (TRMT1) cleavage fragments exhibit alterations in RNA binding and tRNA modification activity. (**A**) Schematic of wild-type TRMT1 and predicted TRMT1 fragments resulting from nonstructural protein 5 (Nsp5) cleavage at Q530N. (**B**) Immunoblot analysis of anti-FLAG purifications from human cells expressing vector control, full-length TRMT1, or TRMT1 cleavage fragments fused to the FLAG tag. The immunoblot was probed with anti-FLAG and anti-actin antibodies. (**C**) Nucleic acid stain of RNAs extracted from the indicated input or purified samples after denaturing PAGE. The migration pattern of 5.8 S rRNA (~150 nt), 5 S rRNA (~120 nt), and tRNAs (~70–80 nt) are denoted. (**D**) Immunoblot of TRMT1 expression in either control-wild-type (WT) or TRMT1-knockout (KO) human 293T cell lines. (**E, F**) Representative gel of primer extension assays to monitor the presence of dimethylguanosine (m2,2G) in tRNA-Met-CAU or mt-tRNA-Ile-GAU from the cell lines transfected with the indicated TRMT1 constructs. D, dihydrouridine; m1G, 1-methylguanosine; >, labeled oligonucleotide used for primer extension. Protein-RNA purification was repeated with comparable results (see source data for repeat).

The online version of this article includes the following source data and figure supplement(s) for figure 5:

**Source data 1.** Raw uncropped immunoblots for *Figure 5*.

**Figure supplement 1.** Confocal microscopy images of 293T cells transiently transfected with constructs expressing tRNA methyltransferase 1 (TRMT1) or TRMT1 fragments fused with green fluorescent protein (GFP).

**Figure supplement 1—source data 1.** Raw uncropped microscopy images for *Figure 5—figure supplement 1*.

TRMT1 or variants in control 239T cells had no detectable effect on m2,2G modification in tRNA-Met-CAU or mt-tRNA-Ile-UAU (*Figure 5E*, lanes 2 through 5).

As expected, the m2,2G modification was absent in tRNA-Met-CAU and mt-tRNA-Ile-UAU isolated from the vector-transfected TRMT1-KO cell line leading to read-through to the next RT block (*Figure 5E*, lane 6). Re-expression of full-length TRMT1 or TRMT1-Q530N in the TRMT1-KO cell line was able to restore m2,2G formation in both tRNA-Met-CAU and mt-tRNA-Ile-UAU (*Figure 5E*, lanes 7 and 9). However, neither the N- nor C-terminal TRMT1 fragment was able to restore m2,2G formation in the tRNAs of the TRMT1-KO cell line (*Figure 5E*, lanes 8 and 10). These results indicate that cleavage of TRMT1 by Nsp5 leads to protein fragments that are inactive for tRNA modification activity.

In addition to TRMT1 activity, we compared the subcellular localization pattern of full-length TRMT1 to the TRMT1 cleavage fragments. To visualize TRMT1, we transiently transfected 293T human cells with constructs expressing human TRMT1 fused at its carboxy terminus to GFP. As a mitochondria marker, we co-expressed a red fluorescent protein fused to the mitochondria targeting signal of pyruvate dehydrogenase (*Figure 5—figure supplement 1*, Mito-RFP). Full-length TRMT1-GFP exhibited cytoplasmic localization along with a punctate signal in the nucleus (*Figure 5—figure supplement 1*, TRMT1-WT, GFP). The fluorescence signal for TRMT1 in the cytoplasm overlapped partially with the RFP-tagged mitochondrial marker (*Figure 5—figure supplement 1*, Merge, yellow signal). The mitochondrial and nuclear localization pattern of full-length TRMT1 is consistent with prior studies demonstrating the presence of a mitochondrial targeting signal and nuclear localization signal in TRMT1 (*Dewe et al., 2017*). The N-terminal TRMT1 fragment exhibited primarily cytoplasmic localization with a greatly reduced signal in the nucleus, consistent with a lack of the nuclear localization signal at the C-terminus (*Figure 5—figure supplement 1*, TRMT1 N-term, GFP, merge). In contrast to the N-terminal TRMT1 fragment, the C-terminal TRMT1 fragment displayed primarily nuclear localization with very little signal in the cytoplasm (*Figure 5—figure supplement 1*, TRMT1 C-term, GFP, merge). These results suggest that TRMT1 in the cytoplasm is the likely target of Nsp5 cleavage with the cleavage fragments displaying altered localization compared to full-length TRMT1.

## TRMT1-deficient human cells exhibit reduced levels of SARS-CoV-2 RNA replication

We next investigated whether TRMT1 expression impacts SARS-CoV-2 replication by infecting the 293T control-wild-type (WT) or TRMT1-KO human cell lines described above with SARS-CoV-2. To render the 293T cell lines permissive for SARS-CoV-2 infection, the cell lines were engineered to stably express the ACE2 receptor from an integrated lentiviral vector (*Figure 6—figure supplement 1A*). Immunoblotting for the SARS-CoV-2 nucleocapsid protein confirmed that SARS-CoV-2 could infect both the control-WT and TRMT1-KO human cell lines expressing ACE2 (*Figure 6—figure supplement 1B*). Using these cell lines, we first monitored the effect of SARS-CoV-2 infection on TRMT1 levels. As expected, there was no detectable TRMT1 in any of the lanes containing lysate from the TRMT1-KO cell line (*Figure 6A*, compare control-WT, lanes 1–3, to TRMT1-KO, lanes 4–6). In the control-WT cells, we detected a decrease in TRMT1 levels at both multiplicity of infection (MOI) of 0.2 and 0.4 (*Figure 6A*, TRMT1, quantified in *Figure 6B*). The decrease in TRMT1 in SARS-CoV-2-infected 293T cells is comparable to the reduction in TRMT1 levels in MRC5-ACE2 human cell lines infected with SARS-CoV-2 (*Figure 1*). We also compared the levels of endogenous TRMT1 after infection with SARS-CoV-2 at a higher MOI of 5.0. Human 293T cells infected at the higher MOI exhibited a reduction in TRMT1 levels to nearly the background of the TRMT1-KO cell lines (*Figure 6C and D*). These results indicate that SARS-CoV-2 infection can induce a drastic reduction in cellular TRMT1 levels.

The ACE2-expressing cell lines were then infected at low or high multiplicity of infection (MOI) followed by harvesting of cells at 24 hr post-infection. Intracellular viral RNA levels were monitored by quantitative RT-PCR of the SARS-CoV-2 envelope protein gene. As expected, titration of SARS-CoV-2 viral particles at either low or high MOI led to a concomitant increase in viral RNA in both control and TRMT1-KO cell lines (*Figure 6E and F*). Notably, TRMT1-KO cells exhibited a ~three to fourfold reduction in viral RNA compared to control cells infected at the same MOI (*Figure 6E and F*). These results suggest that expression of TRMT1 is necessary for efficient SARS-CoV-2 replication in human cells.

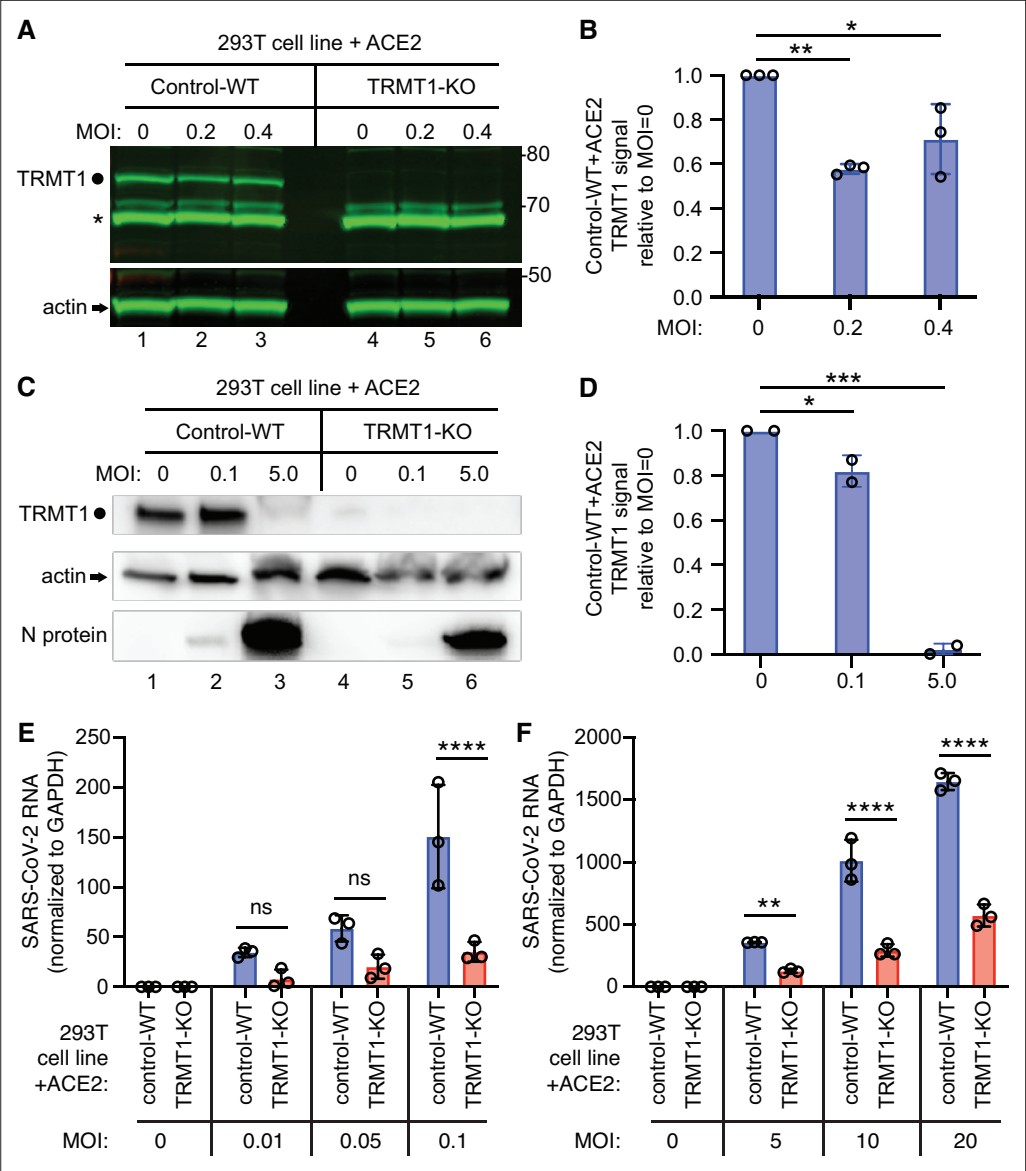

**Figure 6.** The expression of tRNA methyltransferase 1 (TRMT1) affects the levels of severe acute respiratory syndrome coronavirus 2 (SARS-CoV-2) RNA replication in human cells. (**A**) Immunoblot of lysates prepared from 293T control-wild-type (WT) or TRMT1-knockout (KO) cell lines that were mock-infected (multiplicity of infection, MOI of 0) or infected with SARS-CoV-2 at MOI of 0.2 or 0.4 for 24 hr. The immunoblot was probed with antibodies against TRMT1 or actin. Circle represents endogenous full-length TRMT1. Asterisk (*) denotes a non-specific band. Size markers to the right in kiloDalton. (**B**) Normalized TRMT1 signal intensity relative to mock-infected cells (MOI of 0). Statistical significance was determined by one-way ANOVA with Dunnett's multiple comparisons test. (**C**) Immunoblot of lysates prepared from 293T control-wild-type (WT) or TRMT1-KO cell lines that were mock-infected (MOI of 0) or infected with SARS-CoV-2 at MOI of 0.1 or 5.0 for 24 hr. The immunoblot was probed with antibodies as in (**A**). (**D**) Normalized TRMT1 signal intensity relative to mock-infected cells (MOI of 0). (**E, F**) SARS-CoV-2 RNA copy number in control-WT or TRMT1-KO human 293T cell lines after infection at the indicated MOI for 24 hr. Viral copy number was measured by QRT-PCR and normalized to GAPDH. Samples were measured in triplicate. Statistical significance was determined by two-way ANOVA with Šídák's multiple comparisons test. *$p < 0.05$; **$p < 0.01$; ***$p < 0.001$; ****$p < 0.0001$; ns, non-significant.

The online version of this article includes the following source data and figure supplement(s) for figure 6:

**Source data 1.** Raw uncropped immunoblots for *Figure 6*.

**Source data 2.** QRT-PCR measurements of severe acute respiratory syndrome coronavirus 2 (SARS-CoV-2) RNA.

*Figure 6 continued on next page*

*Figure 6 continued*

**Figure supplement 1.** Human 293T cell lines expressing ACE2 can be infected by severe acute respiratory syndrome coronavirus 2 (SARS-CoV-2).

**Figure supplement 1—source data 1.** Raw uncropped immunoblots for *Figure 6—figure supplement 1*.

To confirm that TRMT1-deficiency is the cause of the reduced SARS-CoV-2 replication, we generated TRMT1-KO cell lines re-expressing TRMT1-WT or TRMT1-Q530N with a C-terminal FLAG tag from an integrated lentiviral construct. As expected, wild-type control 293T cells expressed endogenous TRMT1 that was absent in the TRMT1-KO cell lines with empty vector (*Figure 7—figure supplement 1*, TRMT1, lanes 1 and 2). Furthermore, we could detect re-expression of TRMT1-WT or TRMT1-Q530N in the TRMT1-KO cell lines containing the integrated lentiviral TRMT1 expression vectors (*Figure 7—figure supplement 1*, TRMT1, lanes 3–6). We also validated the cell lines by checking for cleavage of the re-expressed TRMT1 by Nsp5. Indeed, the N-terminal TRMT1 cleavage product was detected in the lysate of TRMT1-KO cells expressing TRMT1-WT and Nsp5, but not TRMT1-Q530N and Nsp5 (*Figure 7—figure supplement 1*, TRMT1, compare lanes 3 and 4). These results further support our findings described above that Nsp5 expression leads to site-specific cleavage of TRMT1 in human cells.

We then expressed the ACE2 receptor in the complemented TRMT1-KO cell lines to render them permissive for SARS-CoV-2 infection (*Figure 7—figure supplement 2*). Using these cell lines, we first tested the effect of SARS-CoV-2 infection on TRMT1 levels. Compared to mock-infection, TRMT1 protein levels decreased in the TRMT1-KO cell lines expressing TRMT1-WT after infection with SARS-CoV-2 (*Figure 7A*, lanes 1–3; quantified in 7B). Notably, the level of TRMT1 was not appreciably changed in the TRMT1-KO cell line expressing TRMT1-Q530N after infection with SARS-CoV-2 (*Figure 7A*, lanes 4–6; quantified in 7 C). These results provide evidence that the decrease in endogenous TRMT1 levels after SARS-CoV-2 infection is due to cleavage of TRMT1 by Nsp5.

We next measured SARS-CoV-2 RNA levels after infecting the TRMT1-KO cell lines with SARS-CoV-2. Reproducing our results above, we found that the TRMT1-KO+vector cell line exhibited a decrease in viral RNA levels compared to control cells infected at the same MOI (*Figure 7D*, compare control +vec to KO +vec). TRMT1-KO cell lines re-expressing TRMT1 exhibited higher viral RNA production relative to the TRMT1-KO+vector cell line that was comparable to the control-WT cells (*Figure 7D*, compare KO +vec to KO +WT). Notably, the TRMT1-KO cell line expressing TRMT1-Q530N exhibited a higher level of viral RNA production compared to the TRMT1-KO cell line expressing TRMT1-WT at higher MOI (*Figure 7*, KO +Q530 N). Altogether, these results uncover a requirement for TRMT1 expression for efficient SARS-CoV-2 replication in human cells.

## Impact of TRMT1 on SARS-CoV-2 particle infectivity

Since the presence or absence of TRMT1 appears to impact SARS-CoV-2 RNA production in human 293T cells, we tested if this phenomenon could alter viral particle production. To this end, we collected tissue culture media supernatant containing viral particles generated from control-WT or TRMT1-KO cell lines that were infected by SARS-CoV-2 for 24 hr. The viral titer of the supernatant was determined by focus forming unit assay and particle infectivity expressed relative to viral genomic RNA copy number within the same sample. Consistent with the reduced levels of viral RNA in TRMT1-KO cells infected with SARS-CoV-2, the supernatant from infected TRMT1-KO cells exhibited reduced titer and slightly lower infectivity when compared to viral particles produced from control-WT cells (*Figure 8*, control-WT versus TRMT1-KO). Re-expression of TRMT1 in the TRMT1-KO cells partially restored viral titer at the lower MOI and increased infectivity compared to supernatants from TRMT1-KO cell lines (*Figure 8*, TRMT1-KO+vec versus TRMT1-KO+WT). Interestingly, the expression of the non-cleavable TRMT1-Q530N variant in TRMT1-KO cells promoted an increase of viral titer as well as infectivity compared to expression of wild-type TRMT1 (*Figure 8*, TRMT1-KO+WT versus TRMT1-KO+Q530 N). Altogether, these observations suggest an unexpected role for TRMT1 expression in virus production and the generation of optimally infectious SARS-CoV-2 particles.

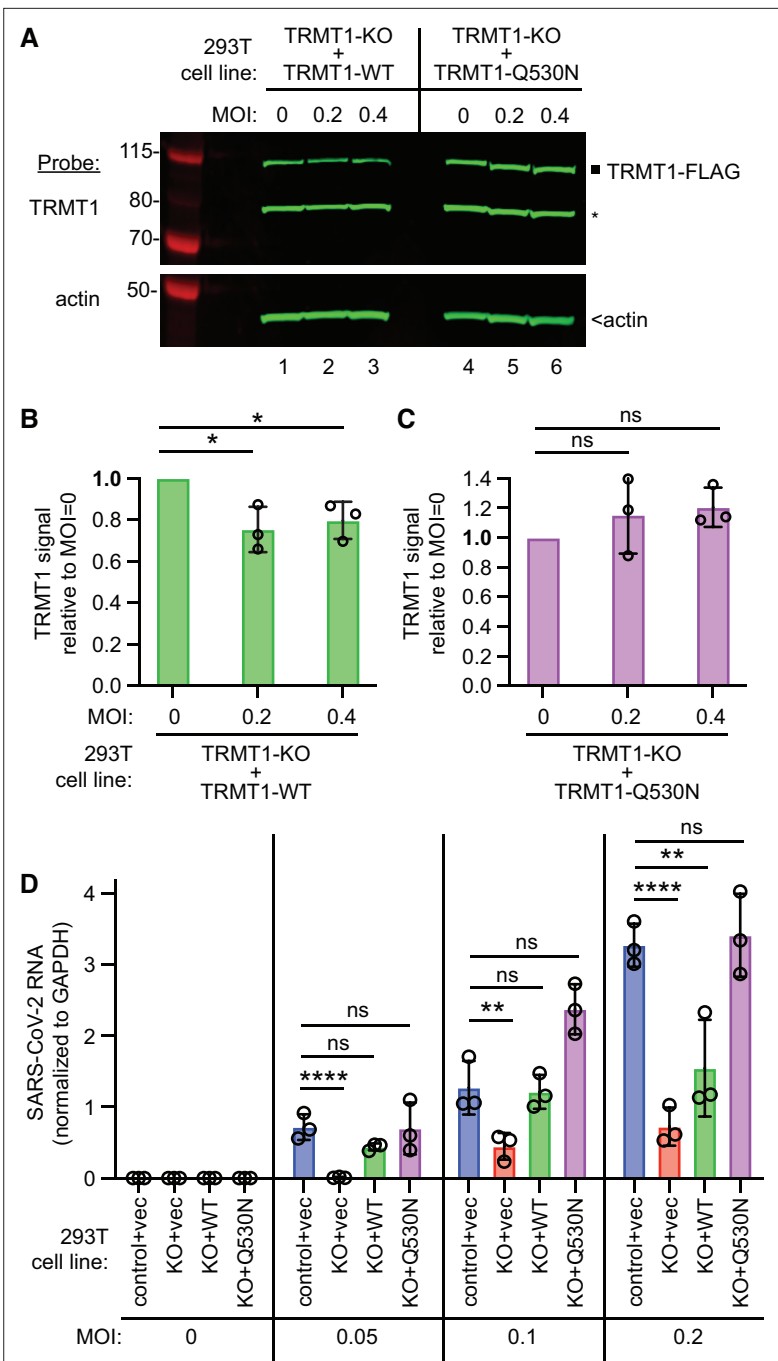

**Figure 7.** tRNA methyltransferase 1 (TRMT1) is required for efficient severe acute respiratory syndrome coronavirus 2 (SARS-CoV-2) replication in human cells. (**A**) Immunoblot of lysates prepared from the indicated 293T TRMT1-knockout (KO) cell lines that were mock-infected (multiplicity of infection, MOI of 0) or infected with SARS-CoV-2 for 24 hr. The immunoblot was probed with antibodies against TRMT1 or actin. Square represents full-length TRMT1-FLAG. Asterisk (*) denotes a non-specific band. Size markers to the left in kiloDalton. (**B**) Normalized TRMT1-WT signal intensity relative to mock-infected cells (MOI of 0). (**C**) Normalized TRMT1-Q530N signal intensity relative to mock-infected cells (MOI of 0). Statistical significance was determined in (**B**) and (**C**) by one-way ANOVA with Dunnett's multiple comparisons test. (**D**) SARS-CoV-2 RNA copy number in control-wild-type (WT) or TRMT1-KO human 293T cell lines after infection at the indicated MOI. Viral copy number was measured by QRT-PCR and normalized to GAPDH. Statistical significance was determined by two-way ANOVA with Dunnett's multiple comparisons test. *p<0.05; **p<0.01; ****p<0.0001; ns, non-significant.

The online version of this article includes the following source data and figure supplement(s) for figure 7:

*Figure 7 continued on next page*

*Figure 7 continued*

**Source data 1.** Raw uncropped immunoblots for *Figure 7A*.

**Source data 2.** QRT-PCR measurements of severe acute respiratory syndrome coronavirus 2 (SARS-CoV-2) RNA for *Figure 7D*.

**Figure supplement 1.** Expression of nonstructural protein 5 (Nsp5) leads to cleavage of tRNA methyltransferase 1 (TRMT1)-wild-type (WT) re-expressed in TRMT1-knockout (KO) cells, but not TRMT1-Q530N.

**Figure supplement 1—source data 1.** Raw uncropped immunoblots for *Figure 7—figure supplement 1*.

**Figure supplement 2.** Immunoblot analysis of lysates prepared from the indicated 293T cell lines expressing empty vector or ACE2.

**Figure supplement 2—source data 1.** Raw uncropped immunoblots for *Figure 7—figure supplement 2*.

## Discussion

Here, we demonstrate that TRMT1 tRNA modification enzyme is an endogenous cleavage target of the SARS-CoV-2 main protease. Our studies parallel the simultaneous work by D'Oliviera et al. that has elucidated how the SARS-CoV-2 main protease recognizes the TRMT1 cleavage sequence (*D Oliviera et al., 2023*). D'Oliviera et al. have determined the structural basis for recognition of TRMT1 by the SARS-CoV-2 main protease, which has revealed a distinct binding mode for certain substrates of the main protease, including TRMT1. Together, our study and the investigation by D'Oliviera et al. provide independent corroboration of each other's main conclusion that TRMT1 is recognized and cleaved by the SARS-CoV-2 main protease in a sequence-specific manner. Moreover, after our manuscript and the manuscript by D'Oliviera et al. were deposited on *BioRxiv*, an independent study by *Lu and Zhou, 2023* has further confirmed our findings that the SARS-CoV-2 main protease can cleave TRMT1.

TRMT1 could represent a coincidental substrate of Nsp5 during SARS-CoV-2 infection due to the presence of a sequence matching the cleavage site of SARS-CoV-2 polyproteins. Since the majority of TRMT1 exhibits steady-state localization to the nucleus and mitochondria of multiple human cell types (*Dewe et al., 2017*), Nsp5 could have access mainly to newly synthesized TRMT1 in the cytoplasm. Moreover, the effect of SARS-CoV-2 infection on TRMT1 levels could depend on the concentration of TRMT1, half-life of TRMT1, and/or amount of Nsp5 expression in the infected cell. This could account for the observation that TRMT1 levels are reduced but not abolished in human cells infected with

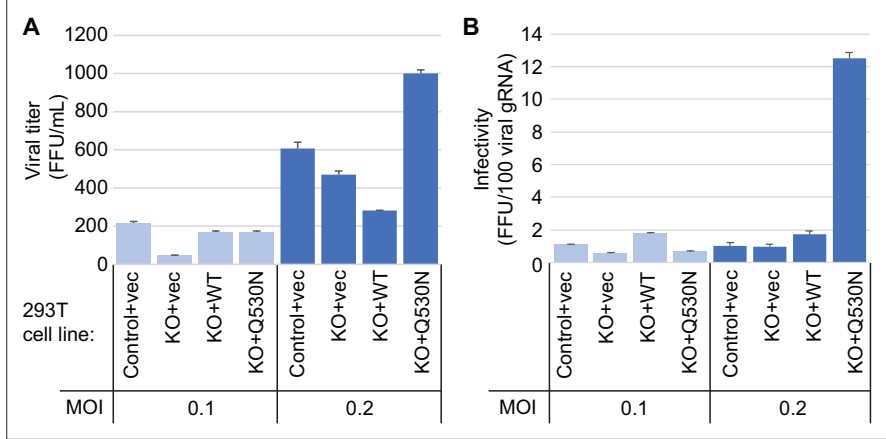

**Figure 8.** Viral infectivity measurement of supernatants collected from the indicated cell lines infected with severe acute respiratory syndrome coronavirus 2 (SARS-CoV-2) for 24 hr. (**A**) Viral titer of supernatants collected from the indicated cell lines infected with SARS-CoV-2. Infectious titer was determined by $TCID_{50}$ endpoint dilution assay in VeroE6 cells and expressed in focus forming units per mL of supernatant (FFU/mL). (**B**) Infectivity of SARS-CoV-2 particles generated from cell lines in (**A**). Infectivity of viral particles was calculated with the formula [(FFU/mL)/(viral genomic RNA copies/mL)], and expressed in FFU per 100 genomic copies.

The online version of this article includes the following source data for figure 8:

**Source data 1.** Infectious titers and QRT-PCR results for severe acute respiratory syndrome coronavirus 2 (SARS-CoV-2) infections.

SARS-CoV-2. Consistent with our results, a subset of proteomic studies has found decreased TRMT1 protein levels in SARS-CoV-2-infected human cells as well as in post-mortem tissue samples from deceased COVID-19 patients (*Bojkova et al., 2020*; *Nie et al., 2021*). These findings suggest that SARS-CoV-2 infection could impact TRMT1 protein levels and tRNA modification patterns within the cells of an infected individual.

While TRMT1 cleavage could be a collateral effect of SARS-CoV-2 infection, the subsequent impact on tRNA modification levels could have physiological consequences on downstream molecular processes that ultimately affect cellular health and/or viral replication. We have previously shown that TRMT1-deficient human cells exhibit decreased levels of global protein synthesis, perturbations in redox metabolism, and reduced proliferation (*Dewe et al., 2017*). Moreover, we have found that partial depletion of TRMT1 is sufficient to increase the oxidative stress sensitivity of human neural stem cells. Thus, the reduction in TRMT1 and TRMT1-catalyzed tRNA modifications observed in human lung cells upon SARS-CoV-2 infection could lead to changes in protein synthesis that affects cellular proliferation and metabolism. In lung tissues, the disruption of TRMT1-dependent processes caused by changes in TRMT1-catalyzed tRNA modification levels could contribute to the pathophysiological outcomes associated with COVID-19 disease. Consistent with this possibility, TRMT1 has been identified as a prognosis factor for SARS-CoV-2 disease severity (*Li et al., 2021b*).

We have found that human cells deficient in TRMT1 display reduced viral RNA levels compared to wild-type cells after SARS-CoV-2 infection. This finding suggests that TRMT1 expression is required for efficient SARS-CoV-2 replication. As mentioned above, TRMT1-deficient human cells exhibit an overall reduction in global protein synthesis due to the loss of m2,2G modifications in tRNAs (*Dewe et al., 2017*). Thus, the TRMT1-deficient human cells could present a cellular environment that is unable to support the levels of translation necessary for efficient virus production. In addition, TRMT1-deficient human cells could exhibit changes in gene expression and cellular metabolism that are suboptimal for SARS-CoV-2 replication.

While TRMT1 could be an unintentional target of Nsp5, it remains conceivable that TRMT1 cleavage modulates the SARS-CoV-2 life cycle through an uncharacterized process. One possibility is that viral RNAs could be substrates of TRMT1-catalyzed RNA methylation. Previous studies have found that endogenous host RNA modification enzymes can modify the genomic and sub-genomic RNAs of SARS-CoV-2 (*Burgess et al., 2021*; *Di Giorgio et al., 2020*; *Li et al., 2021a*; *Peng et al., 2022*; *Zhang et al., 2021a*). Moreover, uncharacterized modifications have been identified through Nanopore sequencing in the genomic and subgenomic RNAs of SARS-CoV-2 (*Kim et al., 2020*). There could be portions of the SARS-CoV-2 genome or subgenomic RNAs that fold into substrates that resemble tRNA targets of TRMT1. There has been precedence for the folding of certain segments of plant viral RNAs into tRNA-like structures that can be modified by cellular tRNA modification enzymes (*Baumstark and Ahlquist, 2001*; *Becker et al., 1998*; *Lesiewicz and Dudock, 1978*). Moreover, the N-terminal TRMT1 fragment could gain the ability to modify viral RNAs due to altered binding specificity since the N-terminal TRMT1 fragment retains the entire methyltransferase domain but not the Zn-finger motif involved in tRNA interaction. Future studies will examine for possible interactions between TRMT1 and the SARS-CoV-2 transcriptome as well as the presence of TRMT1-catalyzed modifications in SARS-CoV-2 RNAs.

We have previously found that TRMT1-deficient human cells exhibit nearly undetectable m2,2G without a major impact on other RNA modifications (*Dewe et al., 2017*). In contrast, human cells infected with SARS-CoV-2 exhibited a decrease in multiple RNA modifications present in a variety of RNAs, including tRNA, rRNA, snRNA, and mRNA. Thus, the widespread changes in RNA modifications after infection with SARS-CoV-2 suggest that coronavirus infection could alter the levels or activity of multiple RNA modification enzymes in addition to TRMT1. Consistent with this hypothesis, RNA modification profiles are changed in response to various forms of cellular stress, including viral infection (*Chan et al., 2018*; *Jungfleisch et al., 2022*). Overall, the study presented here highlights the expanding role of RNA modifications in modulating cellular responses to pathogens that will be the important subject of further investigation.

## Ideas and speculation

Our findings suggest that the SARS-CoV-2 might self-limit its replication by altering the host translation machinery through TRMT1 degradation and reduced levels of m2,2G-modified tRNAs. This

perturbation of the tRNA pool may further inhibit host translation that is already targeted by Nsp1 blockade of mRNA entry on 40 S ribosomes (*Kamitani et al., 2009*; *Lapointe et al., 2021*; *Schubert et al., 2020*; *Thoms et al., 2020*; *Tidu et al., 2020*). The inhibition of host translation may be beneficial to certain viruses that can locally maintain a tRNA pool optimized for viral translation (*Hernandez-Alias et al., 2021*; *Pavon-Eternod et al., 2013*; *van Weringh et al., 2011*; *Yang et al., 2021*). Another possibility is that Nsp5-TRMT1 interaction facilitates the packaging of specific tRNAs into viral particles as suggested previously (*Peña et al., 2022*). The observation that expression of the non-cleavable TRMT1-Q530N variant enhances viral replication and infectivity supports the hypothesis that TRMT1 could facilitate tRNA uptake into viral particles. The packaging of specific tRNAs in viral particles could augment viral translation in the subsequent round of infection, thereby enhancing infectivity and perhaps facilitating the species jump of SARS-CoV-2 towards hosts with incompatible codon bias.

# Materials and methods

**Key resources table**

| Reagent type (species) or resource | Designation | Source or reference | Identifiers | Additional information |
|---|---|---|---|---|
| Cell line (*Homo sapiens*, female) | 293T | ATCC | ATCC: CRL-3216 | |
| Cell line (*Homo sapiens*, male) | MRC5 +ACE2 | *Raymonda et al., 2022* | ATCC: CCL-171 | |
| Strain (coronavirus) | SARS-CoV-2 | BEI resources | NR-52282 | Isolate Hong Kong/VM20001061/2020 |
| Strain (coronavirus) | SARS-CoV-2 | European Virus Archive | 014 V-03890 | Isolate BetaCoV/France/IDF0372/2020 |
| Commercial assay or kit | RNA Clean & Concentrator-5 kit | Zymo | R1013 | |
| Software, algorithm | GraphPad Prism | Dotmatics | Prism 10, version 10.2.2 (341) | |
| Software, algorithm | Chimera | *Pettersen et al., 2004* | X-1.6.1 | |
| Gene (*Homo sapiens*) | *TRMT1* | GenBank | Gene ID: 55621 | |
| Recombinant DNA reagent (plasmid) | pcDNA3.1-TRMT1-FLAG | *Dewe et al., 2017* | | Fu Lab plasmid, mammalian expression vector for wild-type TRMT1 fused to FLAG |
| Recombinant DNA reagent (plasmid) | pcDNA3.1-TRMT1-GFP | *Dewe et al., 2017* | | Fu Lab plasmid, mammalian expression vector for wild-type TRMT1 fused to GFP |
| Recombinant DNA reagent (plasmid) | pcDNA3.1-TRMT1-FLAG-Q530N | This paper | | Fu Lab plasmid, mammalian expression vector for TRMT1-Q530N variant fused to FLAG |
| Recombinant DNA reagent (plasmid) | pcDNA31.-TRMT1-FLAG-N-term | This paper | | Fu Lab plasmid, mammalian expression vector for N-terminal TRMT1 fragment fused to FLAG |
| Recombinant DNA reagent (plasmid) | pcDNA31.-TRMT1-FLAG-C-term | This paper | | Fu Lab plasmid, mammalian expression vector for C-terminal TRMT1 fragment fused to FLAG |
| Recombinant DNA reagent (plasmid) | pLenti-CMV-GFP-Blast | Addgene | Addgene #17445 | |
| Recombinant DNA reagent (plasmid) | pLenti-CMV-TRMT1-FLAG | This paper | | Fu Lab plasmid, lentiviral expression vector for wild-type TRMT1 fused to FLAG |
| Recombinant DNA reagent (plasmid) | pLenti-CMV-TRMT1-FLAG-Q530N | This paper | | Fu Lab plasmid, lentiviral expression vector for TRMT1-Q530N fused to FLAG |
| Recombinant DNA reagent (plasmid) | psPAX2 | Addgene | Addgene # 12260 | |
| Recombinant DNA reagent (plasmid) | pMD2.G | Addgene | Addgene #12259 | |
| Other | MagSTREP 'type3' XT beads, 5% suspension | IBA Lifesciences | 2-4090-002 | For protein purification |
| Other | DYKDDDDK-Tag Monoclonal Antibody Magnetic Microbead | Syd Labs | PA004830 | For protein purification |

*Continued on next page*

*Continued*

| Reagent type (species) or resource | Designation | Source or reference | Identifiers | Additional information |
|---|---|---|---|---|
| Antibody | anti-TRMT1 aa 201–229 (mouse monoclonal) | Santa Cruz Biotechnologies | G3, sc-373687 | Western blot (1:1,000) |
| Antibody | anti-TRMT1 aa 609–659 (rabbit polyclonal) | Bethyl | A304-205A | Western blot (1:500) |
| Antibody | IBA LifeSciences StrepMAB-Classic (mouse monoclonal) | Fisher Scientific | NC9261069 | Western blot (1:1000) |
| Antibody | ANTI-FLAG M2 (Mouse monoclonal) | Sigma | F3165 | Western blot (1:5000) |
| Antibody | anti-SARS-CoV-2 nucleoprotein N protein (Rabbit polyclonal) | Sino Biological | 40068-RP01 | Western blot (1:1000) |
| Antibody | anti-actin C4 (Mouse monoclonal) | EMD Millipore | MAB1501 | Western blot (1:1,000) |
| Antibody | IRDye 800CW anti-mouse IgG (goat polyclonal) | Fisher Scientific | 925–32210 | Western blot secondary (1:10,000) |
| Antibody | IRDye 680RD anti-Mouse IgG (Goat polyclonal) | Li-COR | 925–68070 | Western blot secondary (1:10,000) |
| Other | Odyssey Imager instrument | Li-Cor | CLx | For imaging infrared dye immunoblots. |
| Software, algorithm | Image Studio | Li-Cor | Version 5.2 | |
| Software, algorithm | Fiji (Fiji is just ImageJ) | *Schindelin et al., 2012* | Release 2.15.1 | |
| Recombinant DNA reagent (plasmid) | RRL.sin.cPPT.SFFV/ Ace2.WPRE (MT136) | *Rebendenne et al., 2021* | Addgene 145842 | |
| Antibody | anti-ACE2 Antibody (Goat polyclonal) | R&D systems | AF933 | Western blot (1:200) |
| Sequence-based reagent | SARS_For | This paper | QRT-PCR | ACAGGTACGTTAAT AGTTAATAGCGT |
| Sequence-based reagent | SARS_Rev | This paper | QRT-PCR | ATATTGCAGCAG TACGCACACA |
| Sequence-based reagent | GAPDH_For | This paper | QRT-PCR | GCTCACCGGCAT GGCCTTTCGCGT |
| Sequence-based reagent | GAPDH_Rev | This paper | QRT-PCR | TGGAGGAGTGGG TGTCGCTGTTGA |

## Cell lines

293T and MRC-5 cell lines were obtained from ATCC and authenticated by STR profiling. 293T human embryonic cell lines were cultured in Dulbecco's Minimal Essential Medium (DMEM) containing 10% fetal bovine serum, 2 mM L-alanyl-L-glutamine (GlutaMax, Gibco) and 1% Penicillin/Streptomycin. Cells were grown at 37 °C, 20% Oxygen, and 5% $CO_2$. Telomerase-immortalized MRC5 fibroblasts expressing ACE2 (MRC5-ACE2 cells) were cultured in Dulbecco's modified Eagle serum (DMEM; Invitrogen) supplemented with 10% (vol/vol) fetal bovine serum (FBS) (Atlanta Biologicals), 4.5 g/L glucose, and 1% penicillin-streptomycin (Pen-Strep; Life Technologies) at 37 °C in a 5% (vol/vol) CO2 atmosphere. The cell lines were authenticated by STR DNA profiling and confirmed negative for mycoplasma contamination by Labcorp (https://celllineauthentication.labcorp.com).

## SARS-CoV-2 infection of human cells for protein and RNA analysis

The SARS-CoV-2 isolate, Hong Kong/VM20001061/2020, was previously isolated from a nasopharyngeal aspirate and throat swab from an adult male patient in Hong Kong and was obtained through BEI resources (NR-52282). Viral stocks of SARS-CoV-2 were propagated in Vero-E6 cells in MEM supplemented with 2% (vol/vol) FBS, 4.5 g/L glucose, 1 X Glutamax and 1% penicillin-streptomycin at 37 °C. Viral stock titers were determined by TCID50 analysis in Vero-E6 cells. Experiments involving live SARS-CoV-2 were conducted in a biosafety level 3 facility at the University of Rochester.

For infection, MRC5-ACE2 cells were grown in six-well plates to 90% confluence in growth medium (MRC5-ACE2: DMEM supplemented with 10% FBS and 1% Pen-Strep). Prior to infection, cells were washed with 1 mL of DMEM supplemented with 2% FBS and 1% Pen-Strep. For infection, 750 µL of viral master mix (MOI=5) was added to each well for 1.5 hr. After the adsorption period, the

medium was removed and replaced with fresh DMEM supplemented with 2% FBS and 1% Pen-Strep. To harvest protein, the cell monolayer was washed with 1 mL cold PBS and cells were scraped into 250 µL disruption buffer (250 mM Tris-HCl Ph 7.4, 10% glycerol, 2% β-mercaptoethanol, 5% SDS). DNA was sheared using a sonication probe and samples were stored at –20 °C. To harvest RNA, the medium was removed, and the monolayer was washed with PBS. Cells were collected in 400 µL Trizol and stored at –80 °C.

## Liquid chromatography-mass spectrometry of nucleosides

Total RNA was isolated using Trizol RNA extraction. Small RNAs were subsequently purified from total RNA using the Zymo RNA Clean & Concentrator-5 kit. Small RNAs (1 µg) was digested and analyzed by LC-MS as previously described (*Dewe et al., 2017*; *Zhang et al., 2020*; *Cai et al., 2015*). Briefly, ribonucleosides were separated using a Hypersil GOLD C18 Selectivity Column (Thermo Scientific) followed by nucleoside analysis using a Q Exactive Plus Hybrid Quadrupole-Orbitrap. The modification difference ratio was calculated using the m/z intensity values of each modified nucleoside following normalization to the sum of intensity values for the canonical ribonucleosides; A, U, G, and C. Statistical analysis of the mass spectrometry results was performed using GraphPad Prism software with error bars representing the standard deviation. Statistical tests and the number of times each experiment was repeated are noted in the figure legend. Raw intensity values for each measured nucleoside are provided in the source data file.

## In silico analysis of TRMT1 structure

The Nsp5 cleavage site sequence logo was generated using: https://weblogo.berkeley.edu/logo.cgi.

The predicted tertiary structure of human TRMT1 (Uniprot Q9NXH9) was determined using AlphaFold. Structural visualization was performed using UCSF Chimera software developed by the Resource for Biocomputing, Visualization, and Informatics at the University of California, San Francisco (*Pettersen et al., 2004*). The PDB file for the predicted TRMT1 structures has been provided in the source file.

## Plasmid constructs

The pcDNA3.1-TRMT1-FLAG and pcDNA3.1-TRMT1-GFP expression plasmids have been described previously (*Dewe et al., 2017*). The pcDNA3.1-TRMT1-FLAG-Q530N expression construct was generated by DpnI site-directed mutagenesis. The pcDNA3.1 expression plasmids encoding the N- and C-terminal TRMT1 fragments were generated by PCR cloning. Lentiviral constructs expressing TRMT1-WT or TRMT1-Q530N were generated by T5 Exonuclease DNA Assembly (TEDA) of PCR fragments into pLenti-CMV-GFP-Blast (Addgene 17445) (*Xia et al., 2019*). All plasmid constructs were verified by Sanger sequencing and whole plasmid sequencing (Plasmidsaurus).

## Generation of human cell lines by lentiviral integration

For lentivirus production, 1×10⁶ 293 T cells were seeded onto 60 mm tissue culture dishes. After 24 hr, 1.5 µg of pLenti CMV Blast plasmids containing empty vector or TRMT1 along with 1.5 µg of psPAX2 packaging plasmid and 0.7 µg of pMD2.G envelope plasmid was transfected into the 293T cells using calcium phosphate transfection. Media was changed 16 hr post-transfection. At 48 hr post-transfection, the media containing virus was collected, filter sterilized through a 0.45 µm filter, flash frozen, and stored at –80 °C.

For lentiviral infection of 293T cell lines, 1×10⁶ cells were seeded onto 60 mm tissue culture dishes. After 24 hr, 1 mL of either virus or media for mock infection along with 2 mL of media supplemented with 10 µg/mL of polybrene was added to each well. The cells were washed with PBS and fed fresh media 24 hr post-infection. Polyclonal cell lines with stable integration of the lentiviral constructs were selected with 15 µg/mL blasticidin.

## Transient transfection, protein purification, and immunoblotting

Transient transfection and cellular extract production were performed as previously described (*Fu et al., 2010*). Briefly, 2.5×10⁶ 293 T HEK cells were transiently transfected by calcium phosphate DNA precipitation with 10–20 µg of plasmid DNA. Cells were harvested by trypsinization and washed once with PBS. The cell pellet was resuspended in 500 µL hypotonic lysis buffer (20 mM HEPES, pH 7.9;

2 mM MgCl$_2$; 0.2 mM EGTA, 10% glycerol, 0.1 mM PMSF, 1 mM DTT), incubated on ice for 5 min and subjected to three freeze-thaw cycles in liquid nitrogen and 37 °C. NaCl was added to extracts to a final concentration of 400 mM. After centrifugation at 14,000 x g for 15 min at 4 °C, an equal amount of hypotonic lysis buffer with 0.2% NP-40 was added to 500 μL of soluble cellular extract.

Protein purification from human cell extracts was carried out as previously described with minor adjustments (*Lentini et al., 2020*; *Lentini et al., 2022*). For Strep-tag purification, whole cell extract from transiently transfected cells cell lines (1 mg of total protein) was rotated for 2 hr at 4 ° C in lysis buffer (20 mM HEPES at pH 7.9, 2 mM MgCl2, 0.2 mM EGTA, 10% glycerol, 1 mM DTT, 0.1 mM PMSF, 0.1% NP-40) with 200 mM NaCl. Resin was washed three times using the same buffer followed by protein analysis. Strep-tagged proteins were purified using MagSTREP 'type3' XT beads, 5% suspension (IBA Lifesciences), and eluted with desthiobiotin. FLAG-tagged proteins were purified by incubating whole cell lysates from the transfected cell lines with 20 μL of DYKDDDDK-Tag Monoclonal Antibody Magnetic Microbead (Syd Labs) for 3 hr at 4 °C. Magnetic resin was washed three times in hypotonic lysis buffer with 200 mM NaCl.

Immunoblotting was performed as previously described (*Ramos et al., 2019*). Briefly, cell extracts and purified protein samples were boiled at 95 °C for 5 min followed by fractionation on NuPAGE Bis-Tris polyacrylamide gels (Thermo Scientific). Separated proteins were transferred to Immobilon FL polyvinylidene difluoride (PVDF) membrane (Millipore) for immunoblotting. Membrane was blocked by Odyssey blocking buffer for 1 hrr at room temperature followed by immunoblotting with the following antibodies: mouse monoclonal anti-TRMT1 aa 201–229 (G3, sc-373687, Santa Cruz Biotechnologies), rabbit polyclonal anti-TRMT1 aa 609–659 (Bethyl, A304-205A); anti-Strep-tag II (NC9261069, Thermo Fisher), anti-FLAG epitope tag (L00018; F3165), Rabbit polyclonal anti-SARS-CoV-2 nucleoprotein N protein (40068-RP01; Sino Biological), and anti-actin (L00003; EMD Millipore). Proteins were detected using a 1:10,000 dilution of fluorescent IRDye 800CW goat anti-mouse IgG (925–32210; Thermofisher) or IRDye 680RD Goat anti-Mouse IgG Secondary Antibody (925–68070; Li-COR). Immunoblots were scanned using direct infrared fluorescence via the Odyssey system (LI-COR Biosciences).

Immunoblots were quantified and analyzed using Image Studio Version 5.2 (Li-Cor Biosciences). Rectangles of the same dimension for a given band were used to measure raw signal intensity in either the 700 or 800 nm channel. The intensity of a band in a lane was normalized to actin as a load control. For quantification of the N-terminal TRMT1 cleavage band, we measured the total signal of both the cleavage band and the non-specific band in all lanes. After normalization to actin, the total signal from the cleavage band and the non-specific band in the control lane from cells expressing GFP was subtracted from the lanes with cells expressing Nsp5 to calculate the signal arising from the cleavage band. Statistical analyses were performed using GraphPad Prism software. Where applicable, error bars represent the standard deviation. Statistical tests and the number of times each experiment was repeated are noted in each figure legend. Original files of the full raw unedited blots are available in the accompanying source data file.

## RNA analysis by primer extension

Total RNA was extracted using TRIzol LS reagent (Invitrogen). RNAs were diluted into formamide load buffer, heated to 95 °C for 3 min, and fractionated on a 10% polyacrylamide, Tris-Borate-EDTA (TBE) gel containing 7 M urea. Sybr Gold nucleic acid staining (Invitrogen) was conducted to identify the RNA pattern. For primer extension analysis, 1.5 μg of total RNA was pre-annealed with 5'-$^{32}$P-labeled oligonucleotide and 5 x hybridization buffer (250 mM Tris, pH 8.5, and 300 mM NaCl) in a total volume of 7 μl. The mixture was heated at 95 °C for 3 min followed by slow cooling to 42 °C. An equal amount of extension mix consisting of avian myeloblastosis virus reverse transcriptase (Promega), 5 x AMV buffer, and 40 μM dNTPs was added. The mixture was then incubated at 42 °C for 1 hr and loaded on 15% 7 M urea denaturing polyacrylamide gel. Gels were exposed on a phosphor screen (GE Healthcare) and scanned on a Bio-Rad personal molecular followed by analysis using Fiji NIH ImageJ software (*Schindelin et al., 2012*). Primer extension oligonucleotide sequences were previously described (*Dewe et al., 2017*). Full unedited scan images are available in the accompanying source data file.

## Subcellular localization

For localization of TRMT1 tagged with GFP, 293T human cells were seeded onto coverslips in a six-well plate followed by transfection with the GFP plasmids noted above using Lipofectamine 3000 reagent

(Thermo Fisher). For mitochondrial localization, the cells were infected with baculovirus expressing RFP targeted to mitochondria (CellLight Mitochondria-RFP, BacMam 2.0, Life Technologies). To visualize the nucleus, Hoechst dye was added to the media for 30 min before the cells were washed with PBS, fixed with 4% formaldehyde, and mounted in Aqua Poly/Mount (Polysciences Inc) followed by imaging on a Nikon A1R HD microscope.

## Assays for viral replication

The SARS-CoV-2 was a French Ile de France isolate (https://www.european-virus-archive.com/virus/sars-cov-2-isolate-betacovfranceidf03722020). Viral stocks were generated by amplification on VeroE6 cells. The supernatant was collected, filtered through a 0.45 µm membrane, and tittered using a TCID50 assay. For infections, the cells were previously transduced with a Lentiviral vector expressing ACE2 using the lentiviral construct RRL.sin.cPPT.SFFV/Ace2.WPRE (MT136) was kindly provided by Caroline Goujon (Addgene plasmid # 145842) (*Rebendenne et al., 2021*). Seventy-two hours after transduction, accurate ACE2 expression was controlled on western blot probed with anti-ACE2 antibody (Human ACE2 Antibody, AF933, R&D systems). ACE2-positive cells (70–80% confluence) were then infected with SARS-CoV-2 diluted to achieve the desired MOI. After 24 hr in culture, the cells were lysed with the Luna cell ready lysis module (New England Biolabs).

To measure RNA levels, the amplification reaction was run on a LightcyclerR 480 thermocycler (Roche Diagnostics) using the Luna Universal One-Step RT-qPCR kit (New England Biolabs) with the following primers:

> SARS_For: 5'-ACAGGTACGTTAATAGTTAATAGCGT
> SARS_Rev: 5'-ATATTGCAGCAGTACGCACACA
> GAPDH_For: 5'-GCTCACCGGCATGGCCTTTCGCGT
> GAPDH_Rev: 5'-TGGAGGAGTGGGTGTCGCTGTTGA.

Each qPCR was performed in triplicate, and the means and standard deviations were calculated. Relative quantification of data obtained from RT-qPCR was used to determine changes in SARS-CoV-2 Envelope (E) gene expression across multiple samples after normalization to the internal reference GAPDH gene. The raw qPCR values are provided in the source data file.

## SARS-CoV-2 infectivity measurements

Viral supernatants were collected and filtered on 0.45 µm filters. Virus was then purified by centrifugation through a 20% sucrose cushion at 25,000 rpm for 2.5 hr at 4 °C in a Sw32Ti rotor (Beckman Coulter). Virus pellets were resuspended in PBS and virus genomic RNA copy number was quantified by RTqPCR as described above. Absolute quantification of data obtained from RT-qPCR was used to determine changes in SARS-CoV-2 Envelope (E) gene expression across multiple samples by using external standards of SARS-CoV-2 (dilution $10^7$–10 copies) amplified in parallel within the same instrument run. The raw qPCR values are provided in the source data file. The TCID50 endpoint assay was performed on 96-well plates on VeroE6 cells using the Spearman Karber algorithm (*Cordes et al., 1996*).

## Material Availability

All materials, including plasmids and cell lines, are readily available upon request.

## Acknowledgements

We thank Chenghong Deng, Cailyn Leo, Logan Edvalson, and Sina Ghaemmaghami for comments on this manuscript; the Mass Spectrometry Resource Lab at the University of Rochester; and the Clinical Proteomics Platform, CHU Montpellier.

## Additional information

### Funding

| Funder | Grant reference number | Author |
|---|---|---|
| National Science Foundation | 2033354 | Dragony Fu |
| National Institutes of Health | GM143145 | Dragony Fu |
| National Institutes of Health | GM068411 | Jessica H Ciesla |
| National Institutes of Health | AI1049815 | Jessica H Ciesla |
| National Institutes of Health | AI127370 | Joshua Munger |
| National Institutes of Health | AI50698 | Joshua Munger |
| Montpellier Université d'Excellence | CoVIMOD FRS13 | Patrick Eldin Laurence Briant |

The funders had no role in study design, data collection and interpretation, or the decision to submit the work for publication.

### Author contributions

Kejia Zhang, Patrick Eldin, Conceptualization, Data curation, Formal analysis, Validation, Investigation, Visualization, Methodology, Writing – review and editing; Jessica H Ciesla, Data curation, Investigation, Methodology, Writing – review and editing; Laurence Briant, Conceptualization, Resources, Supervision, Funding acquisition, Methodology, Project administration, Writing – review and editing; Jenna M Lentini, Jillian Ramos, Investigation, Methodology, Writing – review and editing; Justin Cobb, Investigation; Joshua Munger, Resources, Supervision, Funding acquisition, Project administration, Writing – review and editing; Dragony Fu, Conceptualization, Resources, Data curation, Formal analysis, Supervision, Funding acquisition, Validation, Investigation, Visualization, Methodology, Writing - original draft, Project administration, Writing – review and editing

### Author ORCIDs

Kejia Zhang [iD] https://orcid.org/0000-0002-1089-6782
Patrick Eldin [iD] http://orcid.org/0000-0002-3678-1043
Laurence Briant [iD] https://orcid.org/0000-0002-1995-3501
Dragony Fu [iD] https://orcid.org/0000-0002-8725-8658

Reviewer #1 (Public review): https://doi.org/10.7554/eLife.90316.3.sa1
Reviewer #2 (Public review): https://doi.org/10.7554/eLife.90316.3.sa2
Reviewer #3 (Public review): https://doi.org/10.7554/eLife.90316.3.sa3
Author response https://doi.org/10.7554/eLife.90316.3.sa4

## Additional files

### Supplementary files

• MDAR checklist

### Data availability

All data generated or analyzed during this study are included in the manuscript and supporting files.

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
