## [Editor Report · eLife assessment]

This manuscript provides **important** insights into the degradation of the host tRNA modification enzyme TRMT1 by the SARS-CoV-2 protease Nsp5 (nonstructural protein 5 or MPro). The data **convincingly** support the main conclusions of the paper. These results will be of interest to virologists studying the alterations in tRNA modifications, host methyltransferases, and viral infections.

---

## [Referee Report · Reviewer #1 (Public review)]

Zhang et al. investigate the hypothesis that tRNA methyl transferase 1 (TRMT1) is cleaved by NSP5 (nonstructural protein 5 or MPro), the SARS-CoV-2 main protease, during SARS-CoV-2 infection. They provide solid evidence that TRMT1 is a substrate of Nsp5, revealing an Nsp5 target consensus sequence and evidence of TRMT1 cleavage in cells. Their conclusions are exceptionally strong given the co-submission by D'Oliveira et al showing cleavage of TRMT1 in vitro by Nsp5. The detection of the N-terminal TRMT1 fragment by western blot is not robust; however, the authors provide corroborating assays and detailed densitometry methods, providing confidence to this reviewer that a TRMT1 fragment is produced under some conditions. Separately, the authors convincingly demonstrate widespread downregulation of RNA modifications during CoV-2 infection, including a requirement for TRMT1 in efficient viral replication. This finding is congruent with the authors' previous work defining the impact of TRMT1 and m2,2g on global translation, which is most likely necessary to support infection and virion production. Based on the data provided here, TRMT1 cleavage may be an act by CoV-2 to self-limit replication, as expression of a non-cleavable TRMT1 (versus wild type TRMT1) supports enhanced viral RNA expression at certain MOIs. The authors propose a few fascinating ideas to why this may be so in "Ideas and Speculation." Theoretically, TRMT1 cleavage should inactivate the modification activity of TRMT1, which the authors thoroughly and elegantly investigate with rigorous biochemical assays. However, only a minority of TRMT1 undergoes cleavage during infection at low MOIs and thus whether TRMT1 cleavage serves an important functional role during CoV-2 replication will be an important topic for future work. The authors fairly assess their work in this regard. In summary, this study demonstrates an important finding that the tRNA modification landscape is altered during CoV-2 infection, and that TRMT1 is an important host factor supporting CoV-2 replication. Their data pushes forward the idea that control of tRNA expression and functionality is an important and understudied area of host-pathogen interaction.

---

## [Referee Report · Reviewer #2 (Public review)]

Summary:

The manuscript titled 'Proteolytic cleavage and inactivation of the TRMT1 tRNA modification enzyme by SARS-CoV-2 main protease' from K. Zhang et al., demonstrates that several RNA modifications are downregulated during SARS-CoV-2 infection including the widespread m2,2G methylation, which potentially contributes to changes in host translation. To understand the molecular basis behind this global hypomodification of RNA during infection, the authors focused on the human methyltransferase TRMT1 that catalyzes the m2,2G modification. They reveal that TRMT1 not only interacts with the main SARS-CoV-2 protease (Nsp5) in human cells but is also cleaved by Nsp5. To establish if TRMT1 cleavage by Nsp5 contributes to the reduction in m2,2G levels, the authors show compelling evidence that the TRMT1 fragments are incapable of methylating the RNA substrates due to loss of RNA binding by the catalytic domain. They further determine that expression of full-length TRMT1 is required for optimal SARS-CoV-2 replication in 293T cells. Nevertheless, the cleavage of TRMT1 was dispensable for SARS-CoV-2 replication hinting at the possibility that TRMT1 could be an off-target or fortuitous substrate of Nsp5. Overall, this study will be of interest to virologist and biologists studying the role of RNA modification and RNA modifying enzyme in viral infection.

Strengths:

• The authors use state-of-the-art mass spectrometry approach to quantify RNA modifications in human cells infected with SARS-CoV-2.

• The authors go to great lengths to demonstrate that SARS-CoV-2 main protease, Nsp5, interacts and cleaves TRMT1 in cells and perform important controls when needed. They use a series of overexpression with strategically placed tags on both TRMT1 and Nsp5 to strengthen their observations.

• The use of an inactive Nsp5 mutant (C145A) strongly supports the claim of the authors that Nsp5 is solely responsible for TRMT1 cleavage in cells.

• Although the direct cleavage was not experimentally determined, the authors convincingly show that TRMT1 Q530N is not cleaved by Nsp5 suggesting that the predicted cleavage site at this position is most likely the bona fide region processed by Nsp5 in cells.

• To understand the impact of TRMT1 cleavage on its RNA methylation activity, the authors rigorously test four protein constructs for their capacity not only to bind RNA but also to introduce the m2,2G modification. They demonstrate that the fragments resulting from TRMT1 cleavage are inactive and cannot methylate RNA. They further establish that the C-terminal region of TRMT1 (containing a zinc-finger domain) is the main binding site for RNA.

• While 293T cells are unlikely an ideal model system to study SARS-CoV-2 infection, the authors use two cell lines and well-designed rescue experiments to uncover that TRMT1 is required for optimal SARS-CoV-2 replication.

Weaknesses:

• Immunoblotting is extensively used to probe for TRMT1 degradation by Nsp5 in this study. Regretfully, the polyclonal antibody used by the authors shows strong non-specific binding to other epitopes. This complicates the data interpretation and quantification since the cleaved TRMT1 band migrates very closely to a main non-specific band detected by the antibody (for instance Fig 3A). While this reviewer is concerned about the cross-contamination during quantification of the N-TRMT1, the loss of this faint cleaved band with the TRMT1 Q530N mutant is reassuring. Nevertheless, the poor behavior of this antibody for TRMT1 detection was already reported and the authors should have taken better precautions or designed a different strategy to circumvent the limitation of this antibody by relying on additional tags.

• While 293T cells are convenient to use, it is not a well-suited model system to study SARS-CoV-2 infection and replication. Therefore, some of the conclusions from this study might not apply to better suited cell systems such as Vero E6 cells or might not be observed in patient infected cells.

• The reduction of bulk TRMT1 levels is minor during infection of MRC5 cells with SARS-CoV-2 (Fig 1). This does not seem to agree with the more dramatic reduction in m2,2G modification levels. Cellular Localization experiments of TRMT1 would help clarify this. While TRMT1 is found in the cytoplasm and nucleus, it is possible that TRMT1 is more dramatically degraded in the cytoplasm due to easier access by Nsp5.

• In fig 6, the authors show that TRMT1 is required for optimal SARS-CoV-2 replication. This can be rescued by expressing TRMT1 (fig 7). Nevertheless, it is unknown if the methylation activity of TRMT1 is required. The authors could have expressed an inactive TRMT1 mutant (by disrupting the SAM binding site) to establish if the RNA modification by TRMT1 is important for SARS-CoV-2 replication or if it is the protein backbone that might contribute to other processes.

• Fig 7, the authors used the Q530N variant to rescue SARS-CoV-2 replication in TRMT1 KO cells. This is an important experiment and unexpectedly reveals that TRMT1 cleavage by Nsp5 is not required for viral replication. To strengthen the claim of the authors that TRMT1 is required to promote viral replication and that its cleavage inhibits RNA methylation, the authors could express the TRMT1 N-terminal construct in the TRMT1 KO cells to assess if viral replication is restored or not to similar levels as WT TRMT1. This will further validate the potential biological importance of TRMT1 cleavage by Nsp5.

• Fig 7, shows that the TRMT1 Q530N variant rescues SARS-CoV-2 replication to greater levels then WT TRMT1. The authors should discuss this in greater detail and its possible implications with their proposed statement. For instance, are m2,2G levels higher in Q530N compared to WT? Does Q530N co-elute with Nsp5 or is the interaction disrupted in cells?

---

## [Referee Report · Reviewer #3 (Public review)]

Summary:

In this manuscript, the authors have used biochemical approaches to provide compelling evidence for the cleavage of TRMT1 by SARS-CoV-2 Nsp5 protease.

This work is of wide interest to biochemists, cell biologists, and structural biologists in the coronavirus (CoV) field. Furthermore, it substantially advances the understanding of how CoV's interact with host factors during infection and modify cellular metabolism.

Strengths:

The authors provide multiple lines of biochemical evidence to report a TRMT1-Nsp5 interaction during SARS-CoV-2 infection. They show that the host enzyme TRMT1 is cleaved at a specific site, and that it generates fragments that are incapable of functioning properly. This is an important result because TRMT1 is a critical player in host protein synthesis. This also advances our understanding of virus-host interactions during SARS-CoV-2 infections. Furthermore, this revised submission attempts to address the mechanistic role of TRMT1-Nsp5 interaction.

Weaknesses:

The discussion on the enhanced viral infectivity upon expression of the non-cleavable TRMT1 is unclear. As presented, this is a bit contradictory to the suggested function of the TRMT1-Nsp5 interaction in diverting the host tRNA pools towards viral propagation. If the authors' model were correct, then one would expect a non-cleavable TRMT1 to inhibit viral infectivity because the virus would be unable to divert the host tRNA pools towards its propagation. I think this section needs to be written more clearly. But other than this, I have no further questions/suggestions for the authors.

---

## [Author Response]

The following is the authors’ response to the original reviews.

**eLife assessment**
This manuscript provides important insights into the degradation of a host tRNA modification enzyme TRMT1 by SARS-CoV-2 protease nsp5. The data convincingly support the main conclusions of the paper. These results will be of interest to virologists interested in studying the alterations in tRNA modifications, host methyltransferases, and viral infections.
**Public Reviews:**

Response to Public Reviews

We appreciate the reviewers’ assessment that our findings are well supported and provide important insight to the field. We also thank the reviewers for their comments and suggestions that have improved the quality of this manuscript. Through the requested edits and experiments, we provide additional results in this revision that further support and extend our original findings.

We acknowledge the major questions that remain to be addressed, including the biological relevance of TRMT1 cleavage by Nsp5. We note that elucidating the biological role of host protein cleavage by viral proteases has been a long-standing challenge. For example, several endogenous proteins have been identified as cleavage targets of HIV protease, but the functional relevance for many of these cases took decades to resolve or remain unknown to this day. Nonetheless, we have added additional experiments that suggest a possible role for TRMT1 and TRMT1 cleavage in SARS-CoV-2 pathobiology.

Key additions in the revised manuscript include:

• Subcellular localization of full-length TRMT1 and TRMT1 fragments (Supplemental Figure 4).

• Experiments demonstrating that TRMT1 levels are reduced to near background levels in SARS-CoV-2 infected human cells at higher MOI (Figure 6C and D).

• Results showing that expression of the non-cleavable TRMT1 mutant can promote virion particle infectivity (Figure 8).

• The addition of an “Ideas and Speculation” subsection that is now being offered to authors by eLife.

**Reviewer #1 (Public Review):**
Zhang et al. investigate the hypothesis that tRNA methyl transferase 1 (TRMT1) is cleaved by NSP5 (nonstructural protein 5 or MPro), the SARS-CoV-2 main protease, during SARS-CoV-2 infection. They provide solid evidence that TRMT1 is a substrate of Nsp5, revealing an Nsp5 target consensus sequence and evidence of TRMT1 cleavage in cells. Their conclusions are exceptionally strong given the co-submission by D'Oliveira et al showing cleavage of TRMT1 in vitro by Nsp5. Separately, the authors convincingly demonstrate widespread downregulation of RNA modifications during CoV-2 infection, including a requirement for TRMT1 in efficient viral replication. This finding is congruent with the authors' previous work defining the impact of TRMT1 and m2,2g on global translation, which is most likely necessary to support infection and virion production. What still remains unclear is the functional relevance of TRMT1 cleavage by Nsp5 during infection. Based on the data provided here, TRMT1 cleavage may be an act by CoV2 to self-limit replication, as the expression of a non-cleavable TRMT1 (versus wild-type TRMT1) supports enhanced viral RNA expression at certain MOIs. Theoretically, TRMT1 cleavage should inactivate the modification activity of TRMT1, which the authors thoroughly and elegantly investigate with rigorous biochemical assays. However, only a minority of TRMT1 undergoes cleavage during infection in this study and thus whether TRMT1 cleavage serves an important functional role during CoV-2 replication will be an important topic for future work. The authors fairly assess their work in this regard. This study pushes forward the idea that control of tRNA expression and functionality is an important and understudied area of host-pathogen interaction.

We thank the reviewer for the thoughtful assessment of our study.

We acknowledge that only a minority of TRMT1 undergoes cleavage during infection at the originally tested MOI. However, the ~40% reduction in TRMT1 levels after infection with SARS-CoV-2 is quite substantial considering that the TRMT1 in the nucleus and mitochondria are likely to be inaccessible to Nsp5. Moreover, we detected a reduction in m2,2G modification in the infected human cells, providing evidence for a functional impact on TRMT1 activity (Figure 1C).

To further test the effects of SARS-CoV-2 infection on endogenous TRMT1, we infected 293T cells at a higher MOI and measured TRMT1 levels. At MOI=5, we found that SARS-CoV-2 infection led to near complete depletion of TRMT1 in human cells. This result suggests that SARS-CoV-2 infection could have a profound impact on TRMT1 levels during pathogenesis. We have added this new experiment as Figures 6C and D.

Weaknesses noted:The detection of the N-terminal TRMT1 fragment by western blot is not robust. The polyclonal antibody used to detect TRMT1 in this work cross-reacts with a non-specific protein product. Unfortunately, this obstructs the visualization of the predicted N-terminal TRMT1 fragment. It is unclear how the authors were able to perform densitometry, given the interference of the nonspecific band. Additionally, the replicates in the source data make it clear that the appearance of the N-terminal fragment "wisp" under the non-specific band is not seen in every replicate. Though the disappearance of this wisp with mutant Nsp5 and uncleavable TRMT1 is reassuring, the detection of the N-terminal fragment with the TRMT1 antibody should be assessed critically. Considering this group has strong research interests in TRMT1, I assume that attempts to make other antibodies have proved unfruitful. Additionally, N-terminal tagging of TRMT1 is predicted to disrupt the mitochondrial targeting signal, eliminating the potential for using alternative antibodies to see the N-terminal fragment.

We agree that the anti-TRMT1 antibody used here is sub-optimal for detection of the N-terminal TRMT1 fragment. However, as noted by the Reviewer, we provided multiple ways of corroborating that the lower-molecular weight band detected in human cells expressing Nsp5 corresponds to the N-terminal TRMT1 fragment. We have shown that the TRMT1 cleavage band is not detectable in human cells expressing GFP or inactive Nsp5. This indicates that the lower molecular weight TRMT1 band only arises when active Nsp5 protease is expressed. Moreover, the TRMT1 cleavage band is not detectable in TRMT1-KO cell lines, demonstrating that the band arises from TRMT1 cleavage rather than a non-specific protein. We have also detected the C-terminal fragment if TRMT1 is over-expressed with Nsp5. In addition, we have shown that the mutation of the predicted Nsp5 cleavage site in TRMT1 abolishes the appearance of the N- and Cterminal cleavage fragments.

Despite the drawbacks of this antibody, we identified gel running conditions that resolves the non-specific band from the N-terminal TRMT1 cleavage fragment. Thus, for quantification, we measured the total signal of both the cleavage band and the nonspecific band in all lanes (Figure 3). After normalization to actin, the total signal from the cleavage band and the non-specific band in the control lane from cells expressing GFP was subtracted from the lanes with cells expressing Nsp5 to calculate the signal arising from the cleavage band. We have updated our Materials and Methods to provide details on how we quantified the TRMT1 cleavage band.

While we did test other antibodies against TRMT1, none of them were sensitive enough to detect TRMT1 cleavage fragments at endogenous levels. For example, we included results with an antibody targeting the C-terminus of TRMT1 that could not detect TRMT1 cleavage products at endogenous levels (Supplemental Figure 3). However, the antibody could detect the C-terminal TRMT1 fragments if TRMT1 was overexpressed with Nsp5 (Supplemental Figure 3).

These technical issues reiterate the fact that the functional significance of TRMT1 cleavage during CoV-2 infection remains unclear. However, this study demonstrates an important finding that the tRNA modification landscape is altered during CoV-2 infection and that TRMT1 is an important host factor supporting CoV-2 replication.

We agree that the functional relevance of TRMT1 cleavage by Nsp5 remains an open question. Thus, we have added an experiment to test the functional impact of TRMT1 on virion particle production and infectivity (Figure 8). We find that TRMT1 expression is required for optimal virus production, consistent with our observation that TRMT1deficient cells exhibit reduced viral RNA replication. In addition, we find that expression of the non-cleavable TRMT1 mutant can promote virion particle infectivity (Figure 8, TRMT1-Q530N). These results are consistent with the Reviewer’s conclusion that “TRMT1 cleavage may be an act by CoV-2 to self-limit replication, as the expression of a non-cleavable TRMT1 (versus wild-type TRMT1) supports enhanced viral RNA expression at certain MOIs”. We discuss the potential implications of this result and their functional relevance in the “Ideas and Speculation” subsection.

**Reviewer #2 (Public Review):**
Summary:The manuscript titled 'Proteolytic cleavage and inactivation of the TRMT1 tRNA modification enzyme by SARS-CoV-2 main protease' from K. Zhang et al. demonstrates that several RNA modifications are downregulated during SARS-CoV-2 infection including the widespread m2,2G methylation, which potentially contributes to changes in host translation. To understand the molecular basis behind this global hypomodification of RNA during infection, the authors focused on the human methyltransferase TRMT1 that catalyzes the m2,2G modification. They reveal that TRMT1 not only interacts with the main SARS-CoV-2 protease (Nsp5) in human cells but is also cleaved by Nsp5. To establish if TRMT1 cleavage by Nsp5 contributes to the reduction in m2,2G levels, the authors show compelling evidence that the TRMT1 fragments are incapable of methylating the RNA substrates due to loss of RNA binding by the catalytic domain. They further determine that expression of full-length TRMT1 is required for optimal SARS-CoV-2 replication in 293T cells. Nevertheless, the cleavage of TRMT1 was dispensable for SARS-CoV-2 replication hinting at the possibility that TRMT1 could be an off-target or fortuitous substrate of Nsp5. Overall, this study will be of interest to virologists and biologists studying the role of RNA modification and RNA modifying enzymes in viral infection.

We thank the reviewer for the thoughtful assessment of our study.

We agree with the possibility that TRMT1 could be a fortuitous substrate of Nsp5 due to the coincidental presence of a Nsp5 cleavage site in TRMT1. As considered in our Discussion section, TRMT1 cleavage could be a collateral effect of SARS-CoV-2 infection. While TRMT1 could be an off-target substrate during viral infection, the subsequent effect on tRNA modification levels could have physiological consequences on downstream processes that affect cellular health. This information could still be useful for understanding the pathophysiological consequences of SARS-CoV-2 infection in tissues.

Strengths:The authors use a state-of-the-art mass spectrometry approach to quantify RNA modifications in human cells infected with SARS-CoV-2.The authors go to great length to demonstrate that SARS-CoV-2 main protease, Nsp5, interacts, and cleaves TRMT1 in cells and perform important controls when needed. They use a series of overexpression with strategically placed tags on both TRMT1 and Nsp5 to strengthen their observations.The use of an inactive Nsp5 mutant (C145A) strongly supports the claim of the authors that Nsp5 is solely responsible for TRMT1 cleavage in cells.Although the direct cleavage was not experimentally determined, the authors convincingly show that TRMT1 Q530N is not cleaved by Nsp5 suggesting that the predicted cleavage site at this position is most likely the bona fide region processed by Nsp5 in cells.To understand the impact of TRMT1 cleavage on its RNA methylation activity, the authors rigorously test four protein constructs for their capacity not only to bind RNA but also to introduce the m2,2G modification. They demonstrate that the fragments resulting from TRMT1 cleavage are inactive and cannot methylate RNA. They further establish that the C-terminal region of TRMT1 (containing a zinc-finger domain) is the main binding site for RNA.While 293T cells are unlikely an ideal model system to study SARS-CoV-2 infection, the authors use two cell lines and well-designed rescue experiments to uncover that TRMT1 is required for optimal SARS-CoV-2 replication.Weaknesses:Immunoblo0ng is extensively used to probe for TRMT1 degradation by Nsp5 in this study. Regretfully, the polyclonal antibody used by the authors shows strong non-specific binding to other epitopes. This complicates the data interpretation and quantification since the cleaved TRMT1 band migrates very closely to a main non-specific band detected by the antibody (for instance Fig 3A). While this reviewer is concerned about the cross-contamination during quantification of the N-TRMT1, the loss of this faint cleaved band with the TRMT1 Q530N mutant is reassuring. Nevertheless, the poor behavior of this antibody for TRMT1 detection was already reported and the authors should have taken better precautions or designed a different strategy to circumvent the limitation of this antibody by relying on additional tags.

We acknowledge the sub-optimal performance of the commercial anti-TRMT1 antibody used in our study. Nevertheless, we have provided multiple lines of evidence indicating that the lower molecular weight band detected using this antibody corresponds to the N-terminal TRMT1 fragment. As noted by the reviewer, we have shown that the lower molecular weight band disappears using the TRMT1-Q530N non-cleavable mutant. The lower molecular weight signal is also absent in TRMT1-KO cell lines expressing Nsp5. Moreover, we have shown that the TRMT1 cleavage band is undetectable in human cells expressing GFP or inactive Nsp5. We have also detected the C-terminal fragment when TRMT1 is over-expressed with Nsp5.

As discussed in the response to Reviewer 1, we did consider alternative approaches for detecting the N-terminal fragment. We thought about tagging TRMT1 at the N-terminus so that we could detect the cleavage band using a different antibody. However, as noted by Reviewer 1, the tagging of TRMT1 at the N-terminus is likely to disrupt the mitochondrial targeting signal and alter the localization of TRMT1. In addition, we spent considerable time and effort testing alternative antibodies against TRMT1. However, none of them were effective at detecting the N- or C-terminal TRMT1 fragments. For example, we included results with a different antibody targeting the C-terminus of TRMT1 that could not detect TRMT1 cleavage products at endogenous levels but could detect them when TRMT1 was overexpressed with Nsp5 (Supplemental Figure 3).

While 293T cells are convenient to use, it is not a well-suited model system to study SARS-CoV2 infection and replication. Therefore, some of the conclusions from this study might not apply to better-suited cell systems such as Vero E6 cells or might not be observed in patient-infected cells.

We acknowledge the potential caveats associated with using 293T human embryonic cells as a system for testing SARS-CoV2 replication. However, we note that 293T cells have been used as a physiological model for discovering and characterizing key aspects of SARS-CoV-2 biology, including viral replication. For example, SARS-CoV-2 has been shown to exhibit significant replication and virion production in 293T cells expressing ACE2 that can be inhibited by known SARS-CoV-2 antiviral compounds:

• https://www.thelancet.com/journals/lanmic/article/PIIS2666-5247(20)300045/fulltext

• https://www.ncbi.nlm.nih.gov/pmc/articles/PMC9444585/

• https://www.science.org/doi/10.1126/sciadv.add3867

• https://www.pnas.org/doi/full/10.1073/pnas.2025866118

293T cells have also been demonstrated to exhibit cytopathic effects upon SARS-CoV-2 infection that are dependent upon the ACE2 receptor and mirror that of infected lung cells in culture and in patient tissues:

• https://www.embopress.org/doi/full/10.15252/embj.2020106267

• https://journals.asm.org/doi/full/10.1128/jvi.00002-22

• https://journals.plos.org/plospathogens/article?id=10.1371/journal.ppat.1009715

• https://www.nature.com/articles/s41559-021-01407-1

In addition to 293T cells, we have demonstrated that infection of MRC5 human pulmonary fibroblast cells with SARS-CoV-2 results in a decrease in TRMT1 levels and m2,2G modification (Figure 1). The reduction in TRMT1 levels in MRC5 cells after SARS-CoV-2 infection is similar to that observed in 293T cells.

The reduction of bulk TRMT1 levels is minor during infection of MRC5 cells with SARS-CoV-2 (Fig 1). This does not seem to agree with the more dramatic reduction in m2,2G modification levels. Cellular Localization experiments of TRMT1 would help clarify this. While TRMT1 is found in the cytoplasm and nucleus, it is possible that TRMT1 is more dramatically degraded in the cytoplasm due to easier access by Nsp5.

We agree that the processing of newly synthesized TRMT1 in the cytoplasm is likely to be the main cause for the reduction of TRMT1 levels in the infected MRC5 cells. Thus, we followed the Reviewer’s suggestion to conduct cellular localization experiments of TRMT1 (Supplemental Figure 4). Through these experiments, we show that full-length TRMT1 exhibits localization to the cytoplasm, mitochondria, and nucleus, consistent with prior findings from our group and others. This result supports the conclusion that cytoplasmic TRMT1 is the likely target of Nsp5 cleavage while TRMT1 in the nucleus and mitochondria are inaccessible to Nsp5. We also note that the decrease in cytoplasmic TRMT1 could account for the reduction in m2,2G modifications if the cytoplasmic pool of TRMT1 is responsible for modifying any exported tRNAs that were not modified in the nucleus.

In Fig 6, the authors show that TRMT1 is required for optimal SARS-CoV-2 replication. This can be rescued by expressing TRMT1 (Fig 7). Nevertheless, it is unknown if the methylation activity of TRMT1 is required. The authors could have expressed an inactive TRMT1 mutant (by disrupting the SAM binding site) to establish if the RNA modification by TRMT1 is important for SARS-CoV-2 replication or if it is the protein backbone that might contribute to other processes.

We agree that it would be interesting to test if the methylation activity of TRMT1 is important for optimal SARS-CoV-2 replication. However, the present study focuses on the cleavage of TRMT1 by Nsp5 and the biological effects of this cleavage. Thus, we feel that generating another human cell line lies outside the scope of this paper and would be an excellent idea for future studies. We thank the reviewer for the proposed experiment.

Fig 7, the authors used the Q530N variant to rescue SARS-CoV-2 replication in TRMT1 KO cells. This is an important experiment and unexpectedly reveals that TRMT1 cleavage by Nsp5 is not required for viral replication. To strengthen the claim of the authors that TRMT1 is required to promote viral replication and that its cleavage inhibits RNA methylation, the authors could express the TRMT1 N-terminal construct in the TRMT1 KO cells to assess if viral replication is restored or not to similar levels as WT TRMT1. This will further validate the potential biological importance of TRMT1 cleavage by Nsp5.

Indeed, we did not expect to find that human cells expressing the TRMT1-Q530N variant exhibit higher levels of viral replication. This suggests that cleavage of TRMT1 is inhibitory for viral replication. To provide further support for this observation, we analyzed the viral titer and infectivity of supernatants derived from human cells expressing wildtype TRMT1 or TRMT1-Q530N. Consistent with our finding that TRMT1-Q530N cells contain more viral RNA, the media supernatants from TRMT1Q530N expressing cells exhibit higher viral titer and infectivity compared to supernatants from TRMT1-KO cells expressing wildtype TRMT1. These results provide additional evidence that TRMT1 is required to promote viral replication. Moreover, these findings suggest that TRMT1 cleavage and reduced protein synthesis could selflimit viral replication. The additional results have been added as Figure 8.

Fig 7 shows that the TRMT1 Q530N variant rescues SARS-CoV-2 replication to greater levels then WT TRMT1. The authors should discuss this in greater detail and its possible implications with their proposed statement. For instance, are m2,2G levels higher in Q530N compared to WT? Does Q530N co-elute with Nsp5 or is the interaction disrupted in cells?

These are excellent points brought up by the Reviewer. As noted above, we have added an additional experiment that tests the functional relevance of TRMT1 expression and cleavage on virion production and infectivity (Figure 8). Moreover, we have followed the Reviewer’s suggestion and discussed the potential implications of these findings in the “Ideas and Speculation” subsection.

**Reviewer #3 (Public Review):**
Summary:In this manuscript, the authors have used biochemical approaches to provide compelling evidence for the cleavage of TRMT1 by SARS-CoV-2 Nsp5 protease. This work is of wide interest to biochemists, cell biologists, and structural biologists in the coronavirus (CoV) field. Furthermore, it substantially advances the understanding of how CoV's interact with host factors during infection and modify cellular metabolism.

We thank the reviewer for the thoughtful assessment of our study.

Strengths:The authors provide multiple lines of biochemical evidence to report a TRMT1-Nsp5 interaction during SARS-CoV-2 infection. They show that the host enzyme TRMT1 is cleaved at a specific site and that it generates fragments that are incapable of functioning properly. This is an important result because TRMT1 is a critical player in host protein synthesis. This also advances our understanding of virus-host interactions during SARS-CoV-2 infections.Weaknesses:The major weakness is the lack of mechanistic insights into TRMT1-Nsp5 interactions. The authors have provided commendable biochemical data on proving the TRMT1-Nsp5 interaction but without clear mechanistic insights into when this interaction takes place in the context of SARS-CoV-2 propagation, what are the functional consequences of this interaction on host biology, and does this somehow benefit the infecting virus? I feel that the authors played it a bit safe despite having access to several reagents and an extremely promising research direction.

We agree that our findings have prompted questions on the mechanistic and functional relevance of TRMT1 cleavage by Nsp5. To begin addressing the latter point, we have included a new experiment testing the impact of TRMT1 expression and cleavage on SARS-CoV-2 virus production and infectivity (Figure 8). We find that TRMT1-deficient cells infected with SARS-CoV-2 exhibit less virion production and the viruses produced are less infectious. Intriguingly, we find that expression of the non-cleavable TRMT1-Q530N variant in TRMT1-KO cells promotes an increase of viral titer as well as infectivity compared to expression of wildtype TRMT1. These results provide evidence for an unexpected role for TRMT1 expression in virus production and the generation of optimally infectious SARS-CoV-2 particles. We discuss the potential implications of this finding in the “Ideas and Speculation” subsection.

We agree that understanding the timing and effects of Nsp5-TRMT1 interaction will be an important area of investigation moving forward. We would like to include additional time points beyond 24- and 48-hours post-infection. However, we have found that the MRC5-ACE2 cells exhibited increased levels of cell death at 72 and 96-hours postinfection that could confound results (Raymonda et al 2022). Moreover, we would like to know how the reduction in m2,2G modifications affects host tRNA biology and translation. However, these experiments involve large-scale methods such as tRNA sequencing and ribosome profiling which are outside the scope of our current studies and will be the subject of future efforts.

We acknowledge the Reviewer’s assessment that we “played it a bit safe” in discussing the functional consequences of Nsp5-TRMT1 interaction. We aimed for a circumspect interpretation of our results and their biological implications, but might have been too cautious in our conclusions. Thus, we have added an “Ideas and Speculation” subsection that discusses possible reasons for how TRMT1 cleavage and interaction with Nsp5 could benefit the virus. We thank the Reviewer for pointing out this issue in our initial manuscript.

**Recommendations for the authors:**

**Reviewer #1 (Recommendations For The Authors):**
Having reviewed an earlier version of this manuscript, I appreciated the recent progress made by the authors. I felt the entire body of work is quite solid and the interpretations are clear and not overstated. One piece of data I thought deserved a sentence or two of discussion was the complementation assay with Q530N TRMT1. This experiment suggests the possibility that cleavage of TRMT1 by Nsp5 may be an act to self-limit replication, although this result could also be due to the elevated levels of Q530N TRMT1 expression compared to WT. I still think it is worthy of discussion. Another thing I would recommend is to include the length of infection by SARS-CoV-2 in the figure legends.

We thank the reviewer for their positive response and constructive comments.

We have followed the Reviewer’s suggestion to further discuss how cleavage of TRMT1 may act to self-limit replication in the “Ideas and Speculation” subsection. We have also included the length of infection by SARS-CoV-2 in the figure legends.

**Reviewer #2 (Recommendations For The Authors)**:In addition to the comments mentioned in the public review, this reviewer encourages the authors to address the following points:Please clarify the rationale behind choosing 24 and 48 hours post-infection as time points for the analyses (Fig 1). One would expect even lower levels of TRMT1 and RNA modification after 72 and 96 hours post-infection.

We chose the 24 and 48-hour time points since we have shown that MRC5 cells exhibit elevated accumulation of viral RNA at these time points (Raymonda et al 2022). However, at 72 and 96-hours post-infection, we have found that the MRC5-ACE2 cells exhibited cytopathic effects indicative of cell death that could confound results. We have included the rationale for these time points in our revised manuscript.

In Supplementary Figure 3, please add in the legend the meaning of the asterisk symbol.

The asterisks denote non-specific bands that are still detectable in the TRMT1-KO cell line. We have updated the Figure Legend and thank the Reviewer for catching this omission.

In Supplementary Figure 3B, there is an intermediate band in lane 3 with C145A when using the antibody 609-659. The authors should clarify what that band is.

The intermediate band in lane 3 (and in lane 6) of Supplemental Figure 3B represents non-specific detection of the Nsp5-C145A variant that exhibits extremely high levels of expression since it cannot self-cleave. We have clarified the identity of the band in the figure legend.

**Reviewer #3 (Recommendations For The Authors):**
I have only minor comments:Although the authors have done a commendable job of providing compelling biochemical evidence of TRMT1 cleavage by Nsp5, it is not clear how this enhances viral infection. The discussion presents the experimental findings and prior publications as a series of correlated observations without clearly specifying the mechanistic benefits of TRMT1 hijacking towards CoV propagation, or even proposing a mechanistic hypothesis to this end.

We agree with the Reviewer that providing a mechanistic hypothesis on how TRMT1 cleavage impacts virus biology will help inform future studies. We have followed the Reviewer’s suggestion and discuss potential mechanisms in the “Ideas and Speculation” subsection.

How do these experiments inform us about the cell biology of SARS-CoV- infections? Does Nsp5-mediated degradation start early in infection? Is the loss of TRMT1 sustained over the course of the infection? Do Nsp5 concentrations or relative amounts correlate with TRMT1 loss during this period? For instance, is there only a modest increase in Nsp5 levels from 24h to 48h? I would suggest adding a few more data points than just 24h and 48h in the cell culture experiments. As the manuscript stands right now, it will be a bit difficult for readers to appreciate the relevance of this study in its present form.

These are excellent questions raised by the Reviewer. The temporal effects of SARSCoV-2 infection on TRMT1 levels will be an important area to dissect moving forward.

As mentioned above, we would like to include additional time points beyond 24- and 48-hours post-infection. However, at 72 and 96-hours post-infection, we have found that the MRC5-ACE2 cells exhibited increased levels of cell death that could confound results.

However, we do observe a correlation between the level of infection and the amount of TRMT1 depletion. In our newly added Figure 6C and 6D, we show that increasing the MOI leads to a concomitant increase in N-protein production that correlates with the amount of TRMT1 depletion. Moreover, we have added additional experiments to explore the biological relevance of our findings in terms of virion particle production and infectivity. We thank the reviewer for these insightful questions that have improved our manuscript and provide a foundation for future studies.

Related to this previous comment: how do the authors rationalize their inference that TRMT1 is essential for SARS-CoV-2 infection, yet it is cleaved during the infection? What seems to be the advantage of this seemingly contradictory but possibly quite intriguing inference?

We acknowledge the paradox that TRMT1 seems to be essential for SARS-CoV-2 replication but is cleaved during the infection. We propose several hypotheses to explain these findings:

Hypothesis 1: TRMT1 could be a bystander target. The loss of TRMT1 expression leads to a decrease in modifications that impacts translation. This decrease in translation capacity of the infected cells would lead to decreased production of viral proteins and reduced viral replication. This could explain why TRMT1-deficient cells exhibit less virus production. This could also account for why the TRMT1-Q530N mutant might produce more virus. In this case, the cleavage of TRMT1 and biological effects on viral replication and virion production are coincidental. However, even if TRMT1 cleavage and inactivation does not impact viral replication or production, it would still be important to know the cellular impacts that contribute to disease pathogenesis.

Hypothesis 2: The slight diminishment of viral replication due to host translation inhibition could outweigh the benefits of shutting down host responses dependent upon protein synthesis. The decrease in TRMT1-catalyzed tRNA modification caused by Nsp5 cleavage could severely inhibit host translation while viral translation can still be maintained through a tRNA pool optimized for viral translation, albeit at a slightly lower rate than if TRMT1 is not cleaved.

Hypotheses 3: The Nsp5-TRMT1 interaction could allow the virus to bind tRNAs that are packaged in viral particles as suggested previously (Pena et al., 2022). The finding that expression of the non-cleavable TRMT1-Q530N variant enhances viral replication and infectivity supports the hypothesis that TRMT1 could facilitate tRNA uptake into viral particles. The packaging of specific tRNAs in viral particles could enhance viral translation in the subsequent round of infection, thereby enhancing infectivity and perhaps facilitating the species jump of SARS-CoV-2 towards hosts with incompatible codon bias.

We have included these hypotheses in the new “Ideas and Speculation” subsection.